# PROVABLE FEDERATED ADVERSARIAL LEARNING VIA MIN-MAX OPTIMIZATION

## ABSTRACT

Federated learning (FL) is a trending training paradigm to utilize decentralized training data. FL allows clients to update model parameters locally for several epochs, then share them to a global model for aggregation. This training paradigm with multi-local step updating before aggregation exposes unique vulnerabilities to adversarial attacks. Adversarial training is a trending method to improve the robustness of neural networks against adversarial perturbations. First, we formulate a *general* form of federated adversarial learning (FAL) that is adapted from adversarial learning in the centralized setting. On the client side of FL training, FAL has an inner loop to optimize an adversarial to generate adversarial samples for adversarial training and an outer loop to update local model parameters. On the server side, FAL aggregates local model updates and broadcast the aggregated model. We design a global training loss to formulate FAL training as a min-max optimization problem. Unlike the convergence analysis in centralized training that relies on the gradient direction, it is significantly harder to analyze the convergence in FAL for three reasons: 1) the complexity of min-max optimization, 2) model not updating in the gradient direction due to the multi-local updates on the client-side before aggregation and 3) inter-client heterogeneity. Further, we address the challenges using appropriate gradient approximation and coupling techniques and present the convergence analysis in the over-parameterized regime. Our main result theoretically shows that the minimal value of loss function under this algorithm can converge to $\epsilon$ small with chosen learning rate and communication rounds. It is noteworthy that our analysis is feasible for non-IID clients.

## 1 INTRODUCTION

Federated learning (FL) is playing an important role nowadays, as it allows different parties or clients to collaboratively train deep learning models without sharing private data. One popular FL paradigm called FedAvg (McMahan et al., 2017) introduces an easy-to-implement distributed learning method without data sharing. Specifically, it requires a central server to aggregate model updates computed by the local participants (also known as nodes or clients) using local imparticipable private data. Then the central server aggregates these updates to train a globally learned model.

Nowadays deep learning model are exposed to severe threats of adversarial samples. Namely, small adversarial perturbations on the inputs will dramatically change the outputs or output wrong answers (Szegedy et al., 2013). In this regard, much effort has been made to improve neural networks' resistance to such perturbations using adversarial learning (Tramèr et al., 2017; Samangouei et al., 2018; Madry et al., 2018). Among these studies, the adversarial training scheme in Madry et al. (2018) has achieved the good robustness in practice. Madry et al. (2018) proposes an adversarial training scheme that uses projected gradient descent (PGD) to generate alternative adversarial samples as the augmented training set. Generating adversarial examples during neural network training is considered as one of the most effective approaches for adversarial training up to now according to the literature (Carlini & Wagner, 2017; Athalye et al., 2018; Croce & Hein, 2020).

Although adversarial learning has attracted much attention in the centralized domain, its practice in FL is under-explored (Zizzo et al., 2020). Like training classical deep neural networks that use gradient-based methods, FL paradigms are vulnerable to adversarial samples. Adversarial learning in FL brings multiple open challenges due to FL properties on low convergence rate, application

in non-IID environments, and secure aggregation solutions. Hence applying adversarial learning in an FL paradigm may lead to unstable training loss and a lack of robustness. However, a recent practical work Zizzo et al. (2020) observed that although there exist difficulties of convergence, the federation of adversarial training with suitable hyperparameter settings can achieve adversarial robustness and acceptable performance. Motivated by the empirical results, we want to address the provable property of combining adversarial learning into FL from the theoretical perspective.

This work aims to theoretically study the unexplored convergence challenges that lie in the interaction between adversarial training and FL. To achieve a general understanding, we consider a general form of ***federated adversarial learning (FAL)***, which deploys adversarial training scheme on local clients in the most common FL paradigm, FedAvg (McMahan et al., 2017) system. Specifically, FAL has an inner loop of local updating that generates adversarial samples (i.e., using Madry et al. (2018)) for adversarial training and an outer loop to update local model weights on the client side. Then global model is aggregated using FedAvg (McMahan et al., 2017). The algorithm is detailed in Algorithm 1.

We are interested in theoretically understanding the proposed FAL scheme from the aspects of model robustness and convergence:

> *Can federated adversarial learning fit training data robustly and converge with a feasibly sized neural network?*

The theoretical convergence analysis of adversarial training itself is challenging in the centralized training setting. Tu et al. (2018) recently proposed a general theoretical method to analyze the risk bound with adversaries but did not address the convergence problem. The investigation of convergence on over-parameterized neural network has achieved tremendous progress (Du et al., 2019a; Allen-Zhu et al., 2019b;c; Du et al., 2019b; Arora et al., 2019b). The basic statement is that training can converge to sufficiently small training loss in polynomial iterations using gradient descent or stochastic gradient descent when the width of the network is polynomial in the number of training examples when initialized randomly. Recent theoretical analysis (Gao et al., 2019; Zhang et al., 2020b) extends these standard training convergence results to adversarial training settings. To answer the above interesting but challenging question, we formulate FAL as an min-max optimization problem. We extend the convergence analysis on the general formulation of over-parameterized neural networks in the FL setting that allows each client to perform min-max training and generate adversarial examples (see Algorithm 1). Involved challenges are arising in FL convergence analysis due to its unique optimization method: 1) unlike classical centralized setting, the global model of FL does not update in the gradient direction; 2) inter-client heterogeneity issue needs to be considered.

Despite the challenges, we give an affirmative answer to the above question. To the best of our knowledge, this work is the first theoretical study that examines these unexplored challenges on the convergence of adversarial training with FL. The contributions of this paper are:

- We propose a framework to analyze a general form of FAL in over-parameterized neural networks. We follow a natural and valid assumption of data separability that the training dataset are well separated apropos of the adversarial perturbations' magnitude. After sufficient rounds of global communication and certain steps of local gradient descent for each $t$, we obtain the minimal loss close to zero. Notably, our assumptions do not rely on data distribution. Thus the proposed analysis framework is feasible for non-IID clients.

- We are the first to theoretically formulate the convergence of the FAL problem into a min-max optimization framework with the proposed loss function. In FL, the update in the global model is no longer directly determined by the gradient directions due to multiple local steps. To tackle the challenges, we define a new 'gradient', *FL gradient*. With valid ReLU Lipschitz and over-parameterized assumptions, we use gradient coupling for gradient updates in FL to show the model updates of each global updating is bounded in federated adversarial learning.

**Roadmap** The rest of the paper is organized as follows. We overview the related work on federated learning and adversarial training in Section 2. We describe the problem formulation for the federated adversarial learning algorithm and introduce the notations, assumptions to be used, and conditions to be considered to ensure convergence in Section 3. Our main convergence result is presented in

Section 4. In Section 5, we present the core techniques and an overview of the proof. We summarize and conclude our results in Section 6.

## 2 RELATED WORK

**Federated Learning** A efficient and privacy-preserving way to learn from the distributed data collected on the edge devices (a.k.a clients) would be FL. FedAvg is a easy-to-implement distributed learning strategy by aggregating local model updates on the server side and transmitting the averaged model back to local clients. Later, many FL methods are developed baed on FedAvg. Theses FL schemes can be divided into aggregation schemes (McMahan et al., 2017; Wang et al., 2020; Li et al., 2021) and optimization schemes (Reddi et al., 2020; Zhang et al., 2020a). Nearly all the them have the common characteristics that client model are updating using gradient descent-based methods, which is venerable to adversarial attacks. In addition, data heterogeneity brings in huge challeng in FL. For IID data, FL has been proven effective. However, in practice, data mostly distribute as non-IID. Non-IID data could substantially degrade the performance of FL models (Zhao et al., 2018; Li et al., 2019; 2021; 2020a). Despite the potential risk in security and unstable performance in non-IID setting, as FL mitigates the concern of data sharing, it is still a popular and practical solution for distributed data learning in many real applications, such as healthcare (Li et al., 2020b; Rieke et al., 2020), autonomous driving (Liang et al., 2019), IoTs (Wang et al., 2019; Lim et al., 2020).

**Learning with Adversaries** Since the discovery of adversarial examples (Szegedy et al., 2013), to make neural networks robust to perturbations, efforts have been made to propose more effective defense methods. As adversarial examples are an issue of robustness, the popular scheme is to include learning with adversarial examples, which can be traced back to (Goodfellow et al., 2014). It produces adversarial examples and injecting them into training data. Later, Madry et al. (Madry et al., 2018) proposed training on multi-step PGD adversaries and empirically observed that adversarial training consistently achieves small and robust training loss in wide neural networks.

**Federated Adversarial Learning** Adversarial examples, which may not be visually distinguishable from benign samples, are often classified. This poses potential security threats for practical machine learning applications. Adversarial training (Goodfellow et al., 2014; Kurakin et al., 2016) is a popular protocol to train more adversarial robust models by inserting adversarial examples in training. The use of adversarial training in FL presents a number of open challenges, including poor convergence due to multiple local update steps, instability and heterogeneity of clients, cost and security request of communication, and so on. To defend the adversarial attacks in federated learning, limited recent studies have proposed to include adversarial training on clients in the local training steps (Bhagoji et al., 2019; Zizzo et al., 2020). These two works empirically showed the performance of adversarial training. The theoretical analysis of convergence is under explored. In addition, (Zhang et al., 2021) proposed an adversarial training strategy in classical distributed setting, not meeting the specialty in FL.

**Convergence via Over-parameterization** Convergence analysis on over-parameterized neural networks falls in two lines. In the first line of work (Li & Liang, 2018; Allen-Zhu et al., 2019b;c;a), data separability plays a crucial role in deep learning theory, especially in showing the convergence result of over-parameterized neural network training. Denote $\delta$ as the minimum distance between all pairs of data points. Data separability theory requires the width ($m$) of the neural network is at least polynomial factor of all the parameters (i.e. $m \geq \text{poly}(n, d, 1/\delta)$), where $n$ is the number of data points and $d$ is the dimension of data. Another line of work (Du et al., 2019b; Arora et al., 2019a;b; Song & Yang, 2019; Lee et al., 2020) builds on neural tangent kernel (Jacot et al., 2018). It requires the minimum eigenvalue ($\lambda$) of the neural tangent kernel to be lower bounded. Our analysis focuses on the former approach based on data separability.

**Robustness of Federated Learning** Previously there were several works that theoretically analyzed the robustness of federated learning under noise. Yin et al. (2018) developed distributed optimization algorithms that were provably robust against arbitrary and potentially adversarial behavior in distributed computing systems, and mainly focused on achieving optimal statistical performance. Reisizadeh et al. (2020) developed a robust federated learning algorithm by considering a structured

affine distribution shift in users' data. Their analysis was built on several assumptions on the loss functions without a direct connection to neural network.

## 3 PROBLEM FORMULATION

To explore the properties of FAL in deep learning, we formulate the problem in over-parameterized neural network regime. We start by presenting the notations and setup required for federated adversarial learning, then we will describe the loss function we use and our FAL algorithm.

The rest of this section is organized as follows. In Section 3.1 we introduce the notations to be used. In Section 3.2 we state the assumptions on the dataset and initialization values. In Section 3.3 we formally describe our FAL algorithm (Algorithm 1) to be investigated.

### 3.1 NOTATIONS

For a vector $x$, we use $\|x\|_p$ to denote its $\ell_p$ norm, in this paper we mainly consider the situation when $p = 1, 2$, or $\infty$. For a matrix $W \in \mathbb{R}^{d \times m}$, we use $W^\top$ to denote its transpose and use $\mathrm{tr}[W]$ to denote its trace. We use $\|W\|_1, \|W\|_2$ and $\|W\|_F$ to denote the entry-wise $\ell_1$ norm, spectral norm and Frobenius norm of $W$ respectively. For each $j \in [m]$, we let $W_j \in \mathbb{R}^d$ be the $j$-th column of $W$. We let $\|W\|_{2,1}$ denotes $\sum_{j=1}^m \|W_j\|_2$ and $\|W\|_{2,\infty}$ denotes $\max_{j \in [m]} \|W_j\|_2$. We denote Gaussian distribution with mean $\mu$ and covariance $\Sigma$ as $\mathcal{N}(\mu, \Sigma)$. We use $\sigma(\cdot)$ to denote the ReLU function $\sigma(z) = \max\{z, 0\}$, and use $\mathbb{1}\{E\}$ to denote the indicator function of event $E$.

### 3.2 PROBLEM SETUP

**Two-layer ReLU network in FAL**  Following recent theoretical work in understanding neural networks training in deep learning (Du et al., 2019b; Arora et al., 2019a;b; Song & Yang, 2019; Lee et al., 2020), in this paper, we focus on a two-layer neural network that has $m$ neurons in the hidden layer, where each neuron is a ReLU activation function. We define the global network as

$$f_U(x) := \sum_{r=1}^m a_r \cdot \sigma(\langle U_r, x \rangle + b_r) \tag{1}$$

and for $c \in [N]$, we define the local network of client $c$ as

$$f_{W_c}(x) := \sum_{r=1}^m a_r \cdot \sigma(\langle W_{c,r}, x \rangle + b_r). \tag{2}$$

Here $U = (U_1, \ldots, U_m) \in \mathbb{R}^{d \times m}$ is the global hidden weight matrix, $W_c = (W_{c,1}, \ldots, W_{c,m}) \in \mathbb{R}^{d \times m}$ is the local hidden weight matrix of client $c$, and $a = (a_1, \ldots, a_m) \in \mathbb{R}^m$ denotes the output weight vector, $b = (b_1, \ldots, b_m) \in \mathbb{R}^m$ denotes the bias vector. During the process of federated adversarial learning, we only update $U$ and $W$, keeping $a$ and $b$ equal to their initialization values, so we can write the global network as $f_U(x)$ and the local network as $f_{W_c}(x)$. For the situation we don't care about the weight matrix, we write $f(x)$ or $f_c(x)$ for short.

Next, we make some standard assumptions regarding our training set.

**Definition 3.1** (Dataset). *There are $N$ clients and $n = NJ$ data in total.[1] Let $\mathcal{S} = \cup_{c \in [N]} \mathcal{S}_c$ where $\mathcal{S}_c = \{(x_{c,1}, y_{c,1}), \ldots, (x_{c,J}, y_{c,J})\} \subseteq \mathbb{R}^d \times \mathbb{R}$ denotes the $J$ training data of client $c$. Without loss of generality, we assume that for all $c \in [N], j \in [J]$, $\|x_{c,j}\|_2 = 1$ and the last coordinate of each point equals to $1/2$, so we consider $\mathcal{X} := \{x \in \mathbb{R}^d : \|x\|_2 = 1, x_d = 1/2\}$. For simplicity, we also assume that for all $c \in [N], j \in [J]$, $|y_{c,j}| \le 1$.[2]*

We now define the initialization for the neural networks.

---

[1]Without loss of generality, we assume that all clients have same number of training data. Our result can be generalized to the setting where each client has a different number of data as the future work.

[2]Our assumptions on data points are reasonable since we can do scale-up. In addition, $l_2$ norm normalization is a typical technique in experiments. Same assumptions also appears in many previous theoretical works like Arora et al. (2019b); Allen-Zhu et al. (2019a;b).

**Definition 3.2** (Initialization). *The initialization of $a \in \mathbb{R}^m, U \in \mathbb{R}^{d \times m}, b \in \mathbb{R}^m$ is $a(0) \in \mathbb{R}^m, U(0) \in \mathbb{R}^{d \times m}, b(0) \in \mathbb{R}^m$. The initialization of client $c$'s local weight matrix $W_c$ is $W_c(0,0) = U(0)$. Here the second term in $W_c$ denotes iteration of local steps.*

- *For each $r \in [m]$, $a_r(0)$ are i.i.d. sampled from $[-1/m^{1/3}, +1/m^{1/3}]$ uniformly.*

- *For each $i \in [d], r \in [m]$, $U_{i,r}(0)$ and $b_r(0)$ are i.i.d. random Gaussians sampled from $\mathcal{N}(0, 1/m)$. Here $U_{i,r}$ means the $(i,r)$-entry of $U$.*

*For each global iteration $t \in [T]$,*

- *For each $c \in [N]$, the initial value of client $c$'s local weight matrix $W_c$ is $W_c(t,0) = U(t)$.*

Next we formulate the adversary model that will be used.

**Definition 3.3** ($\rho$-Bounded adversary). *Let $\mathcal{F}$ denote the function class. An adversary is a mapping $\mathcal{A} : \mathcal{F} \times \mathcal{X} \times \mathbb{R} \to \mathcal{X}$ which denotes the adversarial perturbation. For $\rho > 0$, we define the $\ell_2$ ball as $\mathcal{B}_2(x, \rho) := \{\widetilde{x} \in \mathbb{R}^d : \|\widetilde{x} - x\|_2 \leq \rho\} \cap \mathcal{X}$, we say an adversary $\mathcal{A}$ is $\rho$-**bounded** if it satisfies $\mathcal{A}(f, x, y) \in \mathcal{B}_2(x, \rho)$. Moreover, given $\rho > 0$, we denote the **worst-case** adversary as $\mathcal{A}^* := \arg\max_{\widetilde{x} \in \mathcal{B}_2(x,\rho)} \ell(f(\widetilde{x}), y)$, where $\ell$ is loss function defined in Definition 3.5.*

**Well-separated training sets** In the over-parameterized regime, it is a standard assumption that the training set is well-separated. Since we deal with adversarial perturbations, we require the following $\gamma$-separability, which is a bit stronger.

**Definition 3.4** ($\gamma$-separability). *Let $\gamma \in (0, 1/2), \delta \in (0, 1/2), \rho \in (0, 1/2)$ denote three parameters such that $\gamma \leq \delta \cdot (\delta - 2\rho)$. We say our training set $\mathcal{S} = \cup_{c \in [N]} \mathcal{S}_c = \cup_{c \in [N], j \in [J]} \{(x_{c,j}, y_{c,j})\} \subset \mathbb{R}^d \times \mathbb{R}$ is **globally $\gamma$-separable** w.r.t a $\rho$-bounded adversary, if $\|x_{c_1,j_1} - x_{c_2,j_2}\|_2 \geq \delta$ holds for any $c_1 \neq c_2$ and $j_1 \neq j_2$.*

It is worth noting that, our problem setup does not require the assumption on independent and identically distribution (IID) on data, thus such a formation can be applied to unique challenge of the non-IID setting in FL.

## 3.3 Federated Adversarial Learning

**Algorithm** We focus on a general FAL framework that is adapted from the most common adversarial training in the classical setting on the client. Specifically, we describe the adversarial learning of a local neural network $f_{W_c}$ against an adversary $\mathcal{A}$ that generate adversarial examples during training as shown in Algorithm 1. As for the analysis of a general theoretical analysis framework, we do not specify the explicit format of $\mathcal{A}$.

The FAL algorithm contains two procedures: one is CLIENTUPDATE running on client side and the other is SERVEREXECUTION running on server side. These two procedures are iteratively processed through communication iterations. Adversarial training is addressed in procedure CLIENTUPDATE. Hence, there are two loops in CLIENTUPDATE procedure: the outer loop is iteration for local model updating; and the inner loop is iteratively generating adversarial samples by the adversary $\mathcal{A}$. In the outer loop in SERVEREXECUTION procedure, the neural network's parameters are updated to reduce its prediction loss on the new adversarial samples.

**Adversary and robust loss** We set the following loss for the sake of technical presentation simplicity, as is customary in prior studies Gao et al. (2019); Allen-Zhu et al. (2019a):

**Definition 3.5** (Lipschitz convex loss). *A loss function $\ell : \mathbb{R} \times \mathbb{R} \to \mathbb{R}$ is said to be a **Lipschitz convex loss**, if it satisfies the following four properties:*

- *non-negative;*

- *convex with respect to the first input of $\ell$;*

- *$1-$Lipshcitz, which means $\|\ell(x_1, y_1) - \ell(x_2, y_2)\|_2 \leq \|(x_1, y_1) - (x_2, y_2)\|_2$;*

- *$\ell(y, y) = 0$ for all $y \in \mathbb{R}$.*

---

**Algorithm 1** Federated Adversarial Learning (FAL)

---

**Notations:** Training sets of clients with each client is indexed by $c$, $\mathcal{S}_c = \{(x_{c,j}, y_{c,j})\}_{j=1}^J$; adversary $\mathcal{A}$; local learning rate $\eta_{\text{local}}$; global learning rate $\eta_{\text{global}}$; local updating iterations $K$; global communication round $T$.

 1: Initialization $a(0) \in \mathbb{R}^m, U(0) \in \mathbb{R}^{d \times m}, b(0) \in \mathbb{R}^m$
 2: For $t = 0 \to T$, we iteratively run **Procedure A** then **Procedure B**
 3: **procedure A**. CLIENTUPDATE$(t, c)$
 4:     $\mathcal{S}_c(t) \leftarrow \emptyset$
 5:     $W_c(t, 0) \leftarrow U(t)$                              ▷ Receive global model weights update.
 6:     **for** $k = 0 \to K - 1$ **do**
 7:         **for** $j = 1 \to J$ **do**
 8:             $\widetilde{x}_{c,j}^{(t)} \leftarrow \mathcal{A}(f_{W_c(t,k)}, x_{c,j}, y_{c,j})$        ▷ Adversarial samples. $f_{W_c}$ is defined as (2).
 9:             $\mathcal{S}_c(t) \leftarrow \mathcal{S}_c(t) \cup (\widetilde{x}_{c,j}^{(t)}, y_{c,j})$
10:         **end for**
11:         $W_c(t, k+1) \leftarrow W_c(t, k) - \eta_{\text{local}} \cdot \nabla_{W_c} \mathcal{L}(f_{W_c(t,k)}, \mathcal{S}_c(t))$
12:     **end for**
13:     $\Delta U_c(t) \leftarrow W_c(t, K) - U(t)$
14:     Send $\Delta U_c(t)$ to SERVEREXECUTION
15: **end procedure**
16: **procedure B**. SERVEREXECUTION$(t)$:
17:     **for** each client $c$ **in parallel do do**
18:         $\Delta U_c(t) \leftarrow$ CLIENTUPDATE$(c, t)$              ▷ Receive local model weights update.
19:         $\Delta U(t) \leftarrow \frac{1}{N} \sum_{c \in [N]} \Delta U_c(t)$
20:         $U(t+1) \leftarrow U(t) + \eta_{\text{global}} \cdot \Delta U(t)$          ▷ Aggregation on the server side.
21:         Send $U(t+1)$ to client $c$ for CLIENTUPDATE$(c, t)$
22:     **end for**
23: **end procedure**

---

In this paper we assume $\ell$ is a Lipschitz convex loss. Next, we define our robust loss function of a neural network, which is based on the adversarial examples generated by a $\rho$-bounded adversary $\mathcal{A}$.

**Definition 3.6** (Training loss). *Given a client's training set $\mathcal{S}_c = \{(x_{c,j}, y_{c,j})\}_{j=1}^J \subset \mathbb{R}^d \times \mathbb{R}$ of $J$ examples, the standard training loss of a neural net $f_c : \mathbb{R}^d \to \mathbb{R}$ is defined as $\mathcal{L}(f_c, S_c) := \frac{1}{J} \sum_{j=1}^J \ell(f_c(x_{c,j}), y_{c,j})$. Given $\mathcal{S} = \cup_{c \in [N]} \mathcal{S}_c$, we define the global loss as $\mathcal{L}(f_U, S) := \frac{1}{NJ} \sum_{c=1}^N \sum_{j=1}^J \ell(f_U(x_{c,j}), y_{c,j})$. Given a $\rho$-bounded adversary $\mathcal{A}$, we define the global loss with respect to $\mathcal{A}$ as*

$$\mathcal{L}_{\mathcal{A}}(f_U) := \frac{1}{NJ} \sum_{c=1}^N \sum_{j=1}^J \ell(f_U(\mathcal{A}(f_c, x_{c,j}, y_{c,j})), y_{c,j}) = \frac{1}{NJ} \sum_{c=1}^N \sum_{j=1}^J \ell(f_U(\widetilde{x}_{c,j}), y_{c,j})$$

*and also define the **worst-case** global robust loss as*

$$\mathcal{L}_{\mathcal{A}^*}(f_U) := \frac{1}{NJ} \sum_{c=1}^N \sum_{j=1}^J \ell(f_U(\mathcal{A}^*(f_c, x_{c,j}, y_{c,j})), y_{c,j}) = \frac{1}{NJ} \sum_{c=1}^N \sum_{j=1}^J \max_{x_{c,j}^* \in \mathcal{B}_2(x_{c,j}, \rho)} \ell\left(f_U(x_{c,j}^*), y_{c,j}\right).$$

*Moreover, since we deal with pseudo-net which is defined in Definition 5.1, we also define the loss of a pseudo-net as $\mathcal{L}(g_c, \mathcal{S}_c) := \frac{1}{J} \sum_{j=1}^J \ell(g_c(x_{c,j}), y_{c,j})$ and $\mathcal{L}(g_U, \mathcal{S}) := \frac{1}{NJ} \sum_{c=1}^N \sum_{j=1}^J \ell(g_U(x_{c,j}), y_{c,j})$.*

## 4  OUR RESULT

The main result of this work is showing the convergence of FAL algorithm (Algorithm 1) in overparameterized neural networks. Specifically, our defined global training loss (Definition 3.6) converges to a small $\epsilon$ with the chosen communication round $T$, local and global learning rate $\eta_{\text{local}}, \eta_{\text{global}}$. It is plausible to see that we can control $\eta_{\text{local}}$ according to the local update steps $K$ to achieve convergence. We now formally present our main result Theorem 4.1.

**Theorem 4.1** (Federated Adversarial Learning). *Let $c_0 \in (0,1)$ be a fixed constant. Let $N$ denotes the total number of clients and $J$ denotes the number of data points per client. Suppose that our training set $\mathcal{S} = \cup_{c\in[N]}\mathcal{S}_c$ is globally $\gamma$-separable for some $\gamma > 0$. Then, for all $\epsilon \in (0,1)$, there exists $R = \mathrm{poly}((NJ/\epsilon)^{1/\gamma})$ that satisfies: for all $m \geq \mathrm{poly}(d, (NJ/\epsilon)^{1/\gamma})$, for every $K \geq 1$ and $T \geq \mathrm{poly}(R/\epsilon)$, with probability at least $1 - \exp(-\Omega(m^{1/3}))$ over the randomness of $a(0) \in \mathbb{R}^m, U(0) \in \mathbb{R}^{d\times m}, b(0) \in \mathbb{R}^m$, running **federated adversarial learning** (Algorithm 1) with step size choices $\eta_{\mathrm{global}} = 1/\mathrm{poly}(NJ, R, 1/\epsilon)$ and $\eta_{\mathrm{local}} = 1/K$ will output a list of weights $\{U(1), U(2), \cdots, U(T)\} \in \mathbb{R}^{d\times m}$ that satisfy*

$$\min_{t\in[T]} \mathcal{L}_{\mathcal{A}}(f_{U(t)}) \leq \epsilon.$$

## 5 PROOF SKETCH

To handle the min-max objective in FAL, we formulate the optimization of FAL in the framework of *online gradient descent*[3] : at each local step $k$ on the client side, firstly the adversary generates adversarial samples and computes the loss function $\mathcal{L}\left(f_{W_c(t,k)}, \mathcal{S}_c(t)\right)$, then the local client learner takes the fresh loss function and update $W_c(t, k+1) = W_c(t,k) - \eta_{\mathrm{local}} \cdot \nabla_{W_c} \mathcal{L}\left(f_{W_c(t,k)}, \mathcal{S}_c(t)\right)$.

Compared with the centralized setting, the key difficulties in the convergence analysis of FL are induced by multiple local updates on the client side and the updates on both local and global sides. Specifically, local updates are not the standard gradient as the centralized adversarial training when $K \geq 2$. We used $-\Delta U(t)$ in substitution of the real gradient of $U$ to update the value of $U(t)$. This brings in challenges to bound the gradient of the neural networks. Nevertheless, gradient bounding is challenging in adversarial training solely. To this end, we use gradient coupling method twice to solve this core problem: firstly we bound the difference between real gradient and FL gradient (defined below), then we bound the difference between pseudo gradient and real gradient.

### 5.1 EXISTENCE OF SMALL ROBUST LOSS

In this section, we denote $\widetilde{U} = U(0)$ as the initialization of global weights $U$ and denote $U(t)$ as the global weights of communication round $t$. $U^*$ is the value of $U$ after small perturbations from $\widetilde{U}$ which satisfies $\|U^* - \widetilde{U}\|_{2,\infty} \leq R/m^{c_1}$, here $c_1 \in (0,1)$ is a constant (e.g. $c_1 = 2/3$), $m$ is the width of the neural network and $R$ is a parameter. We will specify the concrete value of these parameters later in appendix.

We study the over-parameterized neural nets' well-approximated pseudo-network to learn gradient descent for over-parameterized neural nets whose weights are close to initialization. Pseudo-network can be seen as a linear approximation of our two layer ReLU neural network near initialization, and the introducing of pseudo-network makes the proof more intuitive.

**Definition 5.1** (Pseudo-network). *Given weights $U \in \mathbb{R}^{d\times m}$, $a \in \mathbb{R}^m$ and $b \in \mathbb{R}^m$, for a neural network $f_U(x) = \sum_{r=1}^m a_r \cdot \sigma(\langle U_r, x\rangle + b_r)$, we define the corresponding **pseudo-network** $g_U$ : $\mathbb{R}^d \to \mathbb{R}$ as $g_U(x) := \sum_{r=1}^m a_r \cdot \langle U_r(t) - U_r(0), x\rangle \cdot \mathbb{1}\{\langle U_r(0), x\rangle + b_r \geq 0\}$.*

**Existence of small robust loss** In order to obtain our main result, we first show that we can find a $U^*$ which is close to $U(0)$ and also makes $\mathcal{L}_{\mathcal{A}^*}(f_{U^*})$ sufficiently small. Later in Theorem 5.6 we show that the average of $\mathcal{L}_{\mathcal{A}}(f_{U(t)})$ is dominated by $\mathcal{L}_{\mathcal{A}^*}(f_{U^*})$, thus we can prove the minimum of $\mathcal{L}_{\mathcal{A}}(f_{U(t)})$ is $\epsilon$ small.

**Theorem 5.2** (Existence, informal version of Theorem F.3). *For all $\epsilon \in (0,1)$, there exists $M_0 = \mathrm{poly}(d, (NJ/\epsilon)^{1/\gamma})$ and $R = \mathrm{poly}((NJ/\epsilon)^{1/\gamma})$ such that for every $m \geq M_0$, with high probability there exists $U^* \in \mathbb{R}^{d\times m}$ that satisfies $\|U^* - U(0)\|_{2,\infty} \leq R/m^{c_1}$ and $\mathcal{L}_{\mathcal{A}^*}(f_{U^*}) \leq \epsilon$.*

### 5.2 CONVERGENCE RESULT FOR FEDERATED LEARNING

For ease of presentation, we first describe the notions of local and global gradients in our federated adversarial learning setting.

---

[3]We refer our readers to Hazan (2016) for more details regarding online gradient descent.

**Definition 5.3** (Gradient). *For a local real network $f_{W_c(t,k)}$, we denote its gradient by $\nabla(f_c, t, k) := \nabla_{W_c}\mathcal{L}(f_{W_c(t,k)}, \mathcal{S}_c(t))$. If the corresponding pseudo-network is $g_{W_c(t,k)}$, then denote the pseudo-network gradient by $\nabla(g_c, t, k) := \nabla_{W_c}\mathcal{L}(g_{W_c(t,k)}, \mathcal{S}_c(t))$.*

Now we consider the global network. We define pseudo gradient as $\nabla(g, t) := \nabla_U \mathcal{L}(g_{U(t)}, \mathcal{S}(t))$ and define **FL gradient** as $\widetilde{\nabla}(f, t) := -\frac{1}{N}\Delta U(t)$, which is used in the proof of Theorem 5.6. We present our gradient coupling methods in the following two lemmas.

**Lemma 5.4** (Bounding the difference between real gradient and FL gradient, informal version of Lemma E.4). *With probability at least $1 - \exp(-\Omega(m^{c_0}))$ over the initialization, for iterations $t$ such that $\|U(t) - U(0)\|_{2,\infty} \leq 1/o(m)$, the gradients satisfy $\|\nabla(f, t) - \widetilde{\nabla}(f, t)\|_{2,1} \leq o(m)$.*

**Lemma 5.5** (Bounding the difference between pseudo gradient and real gradient, informal version of Lemma E.5). *With probability at least $1 - \exp(-\Omega(m^{c_0}))$ over the initialization, for iterations $t$ such that $\|U(t) - U(0)\|_{2,\infty} \leq 1/o(m)$, the gradients satisfy $\|\nabla(g, t) - \nabla(f, t)\|_{2,1} \lesssim NJ \cdot o(m)$.*

The above two lemmas are essential in proving Theorem 5.6, which is our convergence result.

**Theorem 5.6** (Convergence result, informal version of Theorem E.3). *For all $\epsilon \in (0, 1)$, $R \geq 1$, there exists an $M = \text{poly}(n, R, 1/\epsilon)$, such that for every $m \geq M$, for every $K \geq 1$, for every $T \geq \text{poly}(R/\epsilon)$, with probability at least $1 - \exp(-\Omega(m^{c_0}))$ over the randomness of $a(0) \in \mathbb{R}^m$, $U(0) \in \mathbb{R}^{d \times m}$, $b(0) \in \mathbb{R}^m$, for all $U^*$ such that $\|U^* - U(0)\|_{2,\infty} \leq R/m^{c_1}$, running Algorithm 1 with setting $\eta_{\text{global}} = 1/\text{poly}(NJ, R, 1/\epsilon)$ and $\eta_{\text{local}} = 1/K$ will output weights $(U(t))_{t=1}^T$ that satisfy $\frac{1}{T}\sum_{t=1}^T \mathcal{L}_{\mathcal{A}}(f_{U(t)}) \leq \mathcal{L}_{\mathcal{A}^*}(f_{U^*}) + \epsilon$.*

In the proof of Theorem 5.6 we first bound the local gradient $\nabla_r(f_c, t, k)$. We consider the pseudo-network and bound $\mathcal{L}(g_{U(t)}, S(t)) - \mathcal{L}(g_{U^*}, S(t)) \leq \alpha(t) + \beta(t) + \gamma(t)$, where $\alpha(t) := \langle \widetilde{\nabla}(f, t), U(t) - U^* \rangle, \beta(t) = \|\nabla(f, t) - \widetilde{\nabla}(f, t)\|_{2,1} \cdot \|U(t) - U^*\|_{2,\infty}$ and $\gamma(t) := \|\nabla(g, t) - \nabla(f, t)\|_{2,1} \cdot \|U(t) - U^*\|_{2,\infty}$. In bounding $\alpha(t)$, we unfold $\|U(t+1) - U^*\|_F^2$ and by rearranging we have

$$\alpha(t) = \frac{\eta_{\text{global}}}{2}\|\Delta U(t)\|_F^2 + \frac{1}{2\eta_{\text{global}}} \cdot (\|U(t) - U^*\|_F^2 - \|U(t+1) - U^*\|_F^2).$$

We bound $\|\Delta U(t)\|_F^2 \leq \eta_{\text{local}}K \cdot o(m)$. By doing summation over $t$, we have

$$\sum_{t=1}^T \alpha(t) = \frac{\eta_{\text{global}}}{2}\sum_{t=1}^T \|\Delta U(t)\|_F^2 + \frac{1}{2\eta_{\text{global}}} \cdot \sum_{t=1}^T (\|U(t) - U^*\|_F^2 - \|U(t+1) - U^*\|_F^2)$$

$$\leq \frac{\eta_{\text{global}}}{2}\sum_{t=1}^T \|\Delta U(t)\|_F^2 + \frac{1}{2\eta_{\text{global}}} \cdot \|U(1) - U^*\|_F^2$$

$$\lesssim \eta_{\text{global}}\eta_{\text{local}}TK \cdot o(m) + \frac{1}{\eta_{\text{global}}}mD_{U^*}^2$$

In bounding $\beta(t)$, we apply Lemma 5.4 and have

$$\beta(t) = \|\nabla(f, t) - \widetilde{\nabla}(f, t)\|_{2,1} \cdot \|U(t) - U^*\|_{2,\infty}$$
$$\lesssim o(m) \cdot \|U(t) - U^*\|_{2,\infty}$$
$$\lesssim o(m) \cdot (\|U(t) - \widetilde{U}\|_{2,\infty} + D_{U^*}).$$

where $D_{U^*} := \|U^* - \widetilde{U}\|_{2,\infty} \leq R/m^{c_1}$. As for the first term, we bound

$$\|U(t) - \widetilde{U}\|_{2,\infty} \leq \eta_{\text{global}}\sum_{\tau=1}^t \|\Delta U(\tau)\|_{2,\infty}$$

$$= \eta_{\text{global}}\sum_{\tau=1}^t \|\frac{\eta_{\text{local}}}{N}\sum_{c=1}^N \sum_{k=0}^{K-1} \nabla(f_c, t, k)\|_{2,\infty}$$

$$\leq \frac{\eta_{\text{global}}\eta_{\text{local}}}{N}\sum_{\tau=1}^t \sum_{c=1}^N \sum_{k=0}^{K-1} \|\nabla(f_c, t, k)\|_{2,\infty}$$

$$\leq \eta_{\text{global}}\eta_{\text{local}}tKm^{-1/3}$$

and have $\beta(t) \lesssim \eta_{\text{global}}\eta_{\text{local}}tK \cdot o(m) + o(m) \cdot D_{U^*}$, then we do summation and obtain

$$\sum_{t=1}^{T} \beta(t) \lesssim \eta_{\text{global}}\eta_{\text{local}}T^2K \cdot o(m) + o(m) \cdot TD_{U^*}.$$

In bounding $\gamma(t)$, we apply Lemma 5.5 and have

$$\gamma(t) = \|\nabla(g,t) - \nabla(f,t)\|_{2,1} \cdot \|U(t) - U^*\|_{2,\infty} \lesssim NJ \cdot o(m) \cdot (\|U(t) - \widetilde{U}\|_{2,\infty} + D_{U^*}).$$

Then we do summation over $t$ and have

$$\sum_{t=1}^{T} \gamma(t) \lesssim \eta_{\text{global}}\eta_{\text{local}}T^2KNJ \cdot o(m) + TNJ \cdot o(m)D_{U^*}$$

Putting it together with our choice of our all parameters (i.e. $\eta_{\text{local}}, \eta_{\text{global}}, R, K, T, m$), we obtain

$$\frac{1}{T}\sum_{t=1}^{T}\mathcal{L}(g_{U(t)}, S(t)) - \frac{1}{T}\sum_{t=1}^{T}\mathcal{L}(g_{U^*}, S(t)) \leq \frac{1}{T}(\sum_{t=1}^{T}\alpha(t) + \sum_{t=1}^{T}\beta(t) + \sum_{t=1}^{T}\gamma(t)) \leq O(\epsilon).$$

From Theorem D.2 in appendix, we have: $\sup_{x\in\mathcal{X}}|f_U(x) - g_U(x)| \leq O(\epsilon)$, and thus,

$$\frac{1}{T}\sum_{t=1}^{T}\mathcal{L}(f_{U(t)}, \mathcal{S}(t)) - \frac{1}{T}\sum_{t=1}^{T}\mathcal{L}(f_{U^*}, \mathcal{S}(t)) \leq O(\epsilon). \tag{3}$$

From the definition of $\mathcal{A}^*$ we have $\mathcal{L}(f_{U^*}, S(t)) \leq \mathcal{L}_{\mathcal{A}^*}(f_{U^*})$. From the definition of loss we have $\mathcal{L}(f_{U(t)}, \mathcal{S}(t)) = \mathcal{L}_{\mathcal{A}}(f_{U(t)})$. Moreover, since Eq. (3) holds for all $\epsilon > 0$, we can replace $O(\epsilon)$ with $\epsilon$. Thus we prove that for $\forall \epsilon > 0$,

$$\frac{1}{T}\sum_{t=1}^{T}\mathcal{L}_{\mathcal{A}}(f_{U(t)}) \leq \mathcal{L}_{\mathcal{A}^*}(f_{U^*}) + \epsilon.$$

**Combining the results** From Theorem 5.2 we obtain $U^*$ that is close to $U(0)$ and makes $\mathcal{L}_{\mathcal{A}^*}(f_{U^*})$ close to zero, from Theorem 5.6 we have that the average of $\mathcal{L}_{\mathcal{A}}(f_{U(t)})$ is dominated by $\mathcal{L}_{\mathcal{A}^*}(f_{U^*})$. By aggregating these two results, we prove that the minimal of $\mathcal{L}_{\mathcal{A}}(f_{U(t)})$ is $\epsilon$ small and finish the proof of our main Theorem 4.1.

## 6 CONCLUSION

We have studied the convergence of a general format of adopting adversarial training in FL setting to improve FL training robustness. We propose the general framework, FAL, which deploys adversarial samples generation-based adversarial training method on the client-side and then aggregate local model using FedAvg (McMahan et al., 2017). In FAL, each client is trained via min-max optimization with inner loop adversarial generation and outer loop loss minimization. To the best of our knowledge, we are the first to present the comprehensive proof of theoretical convergence guarantee for over-parameterized ReLU network on the presented FAL strategy, using gradient descent. Unlike the convergence of adversarial training in classical settings, we consider the updates on both local client and global server sides. Our result indicates that we can control learning rates $\eta_{\text{local}}$ and $\eta_{\text{global}}$ according to the local update steps $K$ and global communication round $T$ to make the minimal loss close to zero. The technical challenges lie in the multiple local update steps and heterogeneous data, leading to the difficulties of convergence. Under ReLU Lipschitz and over-prameterization assumptions, we use gradient coupling methods twice. Together, we show the model updates of each global updating bounded in our federated adversarial learning. Note that we do not require IID assumptions for data distribution. In sum, the proposed FAL formulation and analysis framework can well handle the multi-local updates and non-IID data in FL. Moreover, our framework can be generalized to other FL aggregation methods, such as sketching and selective aggregation.

## ETHICS STATEMENT

Ensuring the security and robustness of the deployed algorithms is of paramount importance in AI algorithms nowadays. Recently, machine learning training has revealed its vulnerability to adversarial attacks and tendency to generate wrong predictions. The tread cannot be underestimated in FL, where the heterogeneity among clients and requirement for efficient communication brings in challenges in stable gradient updates. In this regard, our work theoretically shows the feasibility of FAL. As a theoretical work, we do not involve any human subjects or datasets. Our work has the potential positive impact on the machine learning community.

## REPRODUCIBILITY STATEMENT

We provide the proof details in the appendix to ensure reproducibility.

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

**Roadmap** The appendix is organized as follows. We introduce the probability tools to be used in our proof in Section A. In addition, we introduce the preliminaries in Section B. We present the proof overview in Section C and additional remarks used in the proof sketch in Section D. We show the detailed proof for the convergence in Section E and the detailed proof of existence in Section F correspondingly.

## A PROBABILITY TOOLS

We introduce the probability tools that will be used in our proof. First we show three lemmas about the tail bounds for random scalar variables in Lemma A.1, A.2 and A.3:

**Lemma A.1** (Chernoff bound Chernoff (1952)). *Let $X = \sum_{i=1}^n X_i$, where $X_i = 1$ with probability $p_i$ and $X_i = 0$ with probability $1 - p_i$, and all $X_i$ are independent. Let $\mu = \mathbb{E}[X] = \sum_{i=1}^n p_i$. Then*
*1. $\Pr[X \geq (1 + \delta)\mu] \leq \exp(-\delta^2 \mu/3), \forall \delta > 0$ ;*
*2. $\Pr[X \leq (1 - \delta)\mu] \leq \exp(-\delta^2 \mu/2), \forall 0 < \delta < 1$.*

**Lemma A.2** (Hoeffding bound Hoeffding (1963)). *Let $X_1, \cdots, X_n$ denote $n$ independent bounded variables in $[a_i, b_i]$. Let $X = \sum_{i=1}^n X_i$, then we have*

$$\Pr[|X - \mathbb{E}[X]| \geq t] \leq 2 \exp\left(-\frac{2t^2}{\sum_{i=1}^n (b_i - a_i)^2}\right).$$

**Lemma A.3** (Bernstein inequality Bernstein (1924)). *Let $X_1, \cdots, X_n$ be independent zero-mean random variables. Suppose that $|X_i| \leq M$ almost surely, for all $i$. Then, for all positive $t$,*

$$\Pr\left[\sum_{i=1}^n X_i > t\right] \leq \exp\left(-\frac{t^2/2}{\sum_{j=1}^n \mathbb{E}[X_j^2] + Mt/3}\right).$$

Next, we introduce Lemma A.4 about CDF of Gaussian distributions:

**Lemma A.4** (Anti-concentration of Gaussian distribution). *Let $X \sim \mathcal{N}(0, \sigma^2)$, that is, the probability density function of $X$ is given by $\phi(x) = \frac{1}{\sqrt{2\pi\sigma^2}} e^{-\frac{x^2}{2\sigma^2}}$. Then*

$$\Pr[|X| \leq t] \in \left(\frac{2}{3}\frac{t}{\sigma}, \frac{4}{5}\frac{t}{\sigma}\right).$$

Finally, we introduce Lemma A.5 as a concentration result on random matrices and Claim A.6 about elementary anti-concentration property of Gaussian distribution.

**Lemma A.5** (Matrix Bernstein, Theorem 6.1.1 in Tropp (2015)). *Consider a finite sequence $\{X_1, \cdots, X_m\} \subset \mathbb{R}^{n_1 \times n_2}$ of independent, random matrices with common dimension $n_1 \times n_2$. Assume that*

$$\mathbb{E}[X_i] = 0, \forall i \in [m] \quad \text{and} \quad \|X_i\| \leq M, \forall i \in [m].$$

*Let $Z = \sum_{i=1}^m X_i$. Let $\mathrm{Var}[Z]$ be the matrix variance statistic of sum:*

$$\mathrm{Var}[Z] = \max\left\{\left\|\sum_{i=1}^m \mathbb{E}[X_i X_i^\top]\right\|, \left\|\sum_{i=1}^m \mathbb{E}[X_i^\top X_i]\right\|\right\}.$$

*Then*

$$\mathbb{E}[\|Z\|] \leq (2\mathrm{Var}[Z] \cdot \log(n_1 + n_2))^{1/2} + M \cdot \log(n_1 + n_2)/3.$$

*Furthermore, for all $t \geq 0$,*

$$\Pr[\|Z\| \geq t] \leq (n_1 + n_2) \cdot \exp\left(-\frac{t^2/2}{\mathrm{Var}[Z] + Mt/3}\right).$$

We state a standard probabilistic result for Gaussian,

**Claim A.6.** *Let $u \sim \mathcal{N}(0, I_d)$ and $\beta \sim \mathcal{N}(0, 1)$, which are independent. For all $x \in \mathcal{X}$ and $t \geq 0$,*

$$\Pr[|\langle u, x \rangle + \beta| \leq t] = O(t).$$

# B  PRELIMINARIES

## B.1  NOTATIONS

For a vector $x$, we use $\|x\|_p$ to denote its $\ell_p$ norm, in this paper we mainly consider the situation when $p = 1, 2$, or $\infty$.

For a matrix $W \in \mathbb{R}^{d \times m}$, we use $W^\top$ to denote its transpose and use $\text{tr}[W]$ to denote its trace. We use $\|W\|_1, \|W\|_2$ and $\|W\|_F$ to denote the entry-wise $\ell_1$ norm, spectral norm and Frobenius norm of $W$ respectively. For each $j \in [m]$, we let $W_j \in \mathbb{R}^d$ be the $j$-th column of $W$. We let $\|W\|_{2,1}$ denotes $\sum_{j=1}^m \|W_j\|_2$ and $\|W\|_{2,\infty}$ denotes $\max_{j \in [m]} \|W_j\|_2$. For two matrices $A, B$ with the same dimensions, we denote their Euclidean inner product as $\langle A, B \rangle := \text{tr}[A^\top B]$.

We denote Gaussian distribution with mean $\mu$ and covariance $\Sigma$ as $\mathcal{N}(\mu, \Sigma)$. We use $\sigma(\cdot)$ to denote the ReLU function $\sigma(z) = \max\{z, 0\}$, and use $\mathbb{1}\{E\}$ to denote the indicator function of event $E$.

## B.2  TWO LAYER RELU NEURAL NETWORK AND INITIALIZATION

In this paper, we focus on a two-layer neural network that has $m$ neurons in the hidden layer, where each neuron is a ReLU activation function. We define the global network as

$$f_U(x) := \sum_{r=1}^m a_r \cdot \sigma(\langle U_r, x \rangle + b_r) \tag{4}$$

and for $c \in [N]$, we define the local network of client $c$ as

$$f_{W_c}(x) := \sum_{r=1}^m a_r \cdot \sigma(\langle W_{c,r}, x \rangle + b_r). \tag{5}$$

Here $U = (U_1, \ldots, U_m) \in \mathbb{R}^{d \times m}$ is the global hidden weight matrix, $W_c = (W_{c,1}, \ldots, W_{c,m}) \in \mathbb{R}^{d \times m}$ is the local hidden weight matrix of client $c$, and $a = (a_1, \ldots, a_m) \in \mathbb{R}^m$ denotes the output weight vector, $b = (b_1, \ldots, b_m) \in \mathbb{R}^m$ denotes the bias vector. During the process of federated adversarial learning, for convenience we keep $a$ and $b$ equal to their initialized values and only update $U$ and $W$, so we can write the global network as $f_U(x)$ and the local network as $f_{W_c}(x)$. For the situation we don't care about the weight matrix, we write $f(x)$ or $f_c(x)$ for short. Next, we make some standard assumptions regarding our training set.

**Definition B.1** (Dataset). *There are $N$ clients and $n = NJ$ data in total.[4] Let $\mathcal{S} = \cup_{c \in [N]} \mathcal{S}_c$ where $\mathcal{S}_c = \{(x_{c,1}, y_{c,1}), ..., (x_{c,J}, y_{c,J})\} \subseteq \mathbb{R}^d \times \mathbb{R}$ denotes the $J$ training data of client $c$. Without loss of generality, we assume that for all $c \in [N], j \in [J]$, $\|x_{c,j}\|_2 = 1$ and the last coordinate of each point equals to $1/2$, so we consider $\mathcal{X} := \{x \in \mathbb{R}^d : \|x\|_2 = 1, \ x_d = 1/2\}$. For simplicity, we also assume that for all $c \in [N], j \in [J], |y_{c,j}| \leq 1$.[5]*

We now define the initialization for the neural networks.

**Definition B.2** (Initialization). *The initialization of $a \in \mathbb{R}^m, U \in \mathbb{R}^{d \times m}, b \in \mathbb{R}^m$ is $a(0) \in \mathbb{R}^m, U(0) \in \mathbb{R}^{d \times m}, b(0) \in \mathbb{R}^m$. The initialization of client $c$'s local weight matrix $W_c$ is $W_c(0, 0) = U(0)$. Here the second term in $W_c$ denotes iteration of local steps.*

- *For each $r \in [m]$, $a_r(0)$ are i.i.d. sampled from $[-1/m^{1/3}, +1/m^{1/3}]$[6] uniformly.*

- *For each $i \in [d], r \in [m]$, $U_{i,r}(0)$ and $b_r(0)$ are i.i.d. random Gaussians sampled from $\mathcal{N}(0, 1/m)$. Here $U_{i,r}$ means the $(i, r)$-entry of $U$.*

---

[4]Without loss of generality, we assume that all clients have same number of training data. Our result can be generalized to the setting where each client has a different number of data as the future work.

[5]Our assumptions on data points are reasonable since we can do scale-up. In addition, $l_2$ norm normalization is a typical technique in experiments. Same assumptions also appears in many previous theoretical works like Arora et al. (2019b); Allen-Zhu et al. (2019a;b).

[6]Here the choice of $m^{1/3}$ is not a must. Actually what we need is $[-1/m^c, +1/m^c]$ for some $c$ that satisfies $\Omega(1) \leq c \leq 1/3$.

*For each global iteration $t \in [T]$,*

- *For each $c \in [N]$, the initial value of client $c$'s local weight matrix $W_c$ is $W_c(t,0) = U(t)$.*

## B.3 ADVERSARY AND WELL-SEPARATED TRAINING SETS

We first formulate the adversary as a mapping.

**Definition B.3** ($\rho$-Bounded adversary). *Let $\mathcal{F}$ denote the function class. An adversary is a mapping $\mathcal{A} : \mathcal{F} \times \mathcal{X} \times \mathbb{R} \to \mathcal{X}$ which denotes the adversarial perturbation. For $\rho > 0$, we define the $\ell_2$ ball as $\mathcal{B}_2(x, \rho) := \{\widetilde{x} \in \mathbb{R}^d : \|\widetilde{x} - x\|_2 \leq \rho\} \cap \mathcal{X}$, we say an adversary $\mathcal{A}$ is $\rho$-**bounded** if it satisfies*

$$\mathcal{A}(f, x, y) \in \mathcal{B}_2(x, \rho).$$

*Moreover, given $\rho > 0$, we denote the **worst-case** adversary as $\mathcal{A}^* := \mathrm{argmax}_{\widetilde{x} \in \mathcal{B}_2(x,\rho)} \ell(f(\widetilde{x}), y)$, where $\ell$ is loss function defined in Definition B.5.*

In the over-parameterized regime, it is a standard assumption that the training set is well-separated. Since we deal with adversarial perturbations, we require the following $\gamma$-separability, which is a bit stronger.

**Definition B.4** ($\gamma$-separability). *Let $\gamma \in (0, 1/2), \delta \in (0, 1/2), \rho \in (0, 1/2)$ denote three parameters such that $\gamma \leq \delta \cdot (\delta - 2\rho)$. We say our training set $\mathcal{S} = \cup_{c \in [N]} \mathcal{S}_c = \cup_{c \in [N], j \in [J]} \{(x_{c,j}, y_{c,j})\} \subset \mathbb{R}^d \times \mathbb{R}$ is **globally $\gamma$-separable** w.r.t a $\rho$-bounded adversary, if*

$$\min_{c_1 \neq c_2, j_1 \neq j_2} \|x_{c_1,j_1} - x_{c_2,j_2}\|_2 \geq \delta.$$

It is worth noting that, our problem setup does not require the assumption on independent and identically distribution (IID) on data, thus such a formation can be applied to unique challenge of the non-IID setting in FL.

## B.4 ROBUST LOSS FUNCTION

We define the following Lipschitz convex loss function that will be used.

**Definition B.5** (Lipschitz convex loss). *A loss function $\ell : \mathbb{R} \times \mathbb{R} \to \mathbb{R}$ is said to be a **Lipschitz convex loss**, if it satisfies the following four properties:*

- *non-negative;*

- *convex in the first input of $\ell$;*

- $1-$*Lipshcitz, which means $\|\ell(x_1, y_1) - \ell(x_2, y_2)\|_2 \leq \|(x_1, y_1) - (x_2, y_2)\|_2$;*

- $\ell(y, y) = 0$ *for all $y \in \mathbb{R}$.*

The choice of this type of loss is for the sake of technical presentation simplicity, as is customary in prior studies Gao et al. (2019); Allen-Zhu et al. (2019a).

We assume $\ell$ is a Lipschitz convex loss in this paper. Next, we define our robust loss function of a neural network, which is based on the adversarial examples generated by a $\rho$-bounded adversary $\mathcal{A}$.

**Definition B.6** (Training loss). *Given a client's training set $\mathcal{S}_c = \{(x_{c,j}, y_{c,j})\}_{j=1}^J \subset \mathbb{R}^d \times \mathbb{R}$ of $J$ examples, the standard training loss of a neural net $f_c : \mathbb{R}^d \to \mathbb{R}$ is defined as $\mathcal{L}(f_c, \mathcal{S}_c) := \frac{1}{J} \sum_{j=1}^J \ell(f_c(x_{c,j}), y_{c,j})$. Given $\mathcal{S} = \cup_{c \in [N]} \mathcal{S}_c$, the global loss is defined as $\mathcal{L}(f_U, \mathcal{S}) := \frac{1}{NJ} \sum_{c=1}^N \sum_{j=1}^J \ell(f_U(x_{c,j}), y_{c,j})$. Given a $\rho$-bounded adversary $\mathcal{A}$, we define the global loss with respect to $\mathcal{A}$ as*

$$\mathcal{L}_{\mathcal{A}}(f_U) := \frac{1}{NJ} \sum_{c=1}^N \sum_{j=1}^J \ell(f_U(\mathcal{A}(f_c, x_{c,j}, y_{c,j})), y_{c,j})$$

$$= \frac{1}{NJ} \sum_{c=1}^N \sum_{j=1}^J \ell(f_U(\widetilde{x}_{c,j}), y_{c,j})$$

*and also define the **worst-case** global robust loss as*

$$\mathcal{L}_{\mathcal{A}^*}(f_U) := \frac{1}{NJ} \sum_{c=1}^{N} \sum_{j=1}^{J} \ell(f_U(\mathcal{A}^*(f_c, x_{c,j}, y_{c,j})), y_{c,j})$$

$$= \frac{1}{NJ} \sum_{c=1}^{N} \sum_{j=1}^{J} \max_{x_{c,j}^* \in \mathcal{B}_2(x_{c,j}, \rho)} \ell\left(f_U(x_{c,j}^*), y_{c,j}\right).$$

*Moreover, since we deal with pseudo-net which is defined in Definition D.1, we also define the loss of a pseudo-net as $\mathcal{L}(g_c, \mathcal{S}_c) := \frac{1}{J} \sum_{j=1}^{J} \ell(g_c(x_{c,j}), y_{c,j})$ and $\mathcal{L}(g_U, \mathcal{S}) := \frac{1}{NJ} \sum_{c=1}^{N} \sum_{j=1}^{J} \ell(g_U(x_{c,j}), y_{c,j})$.*

## B.5 FEDERATED ADVERSARIAL LEARNING ALGORITHM

Classical adversarial training algorithm can be found in Zhang et al. (2020b). Different from the classical setting, our federated adversarial learning of a local neural network $f_{W_c}$ against an adversary $\mathcal{A}$ is shown in Algorithm 2, where there are two procedures: one is CLIENTUPDATE running on client side and the other is SERVEREXECUTION running on server side. These two procedures are iteratively processed through communication iterations. Adversarial training is addressed in procedure CLIENTUPDATE. Hence, there are two loops in CLIENTUPDATE procedure: the outer loop is iteration for local model updating; and the inner loop is iteratively generating adversarial samples by the adversary $\mathcal{A}$. In the outer loop in SERVEREXECUTION procedure, the neural network's parameters are updated to reduce its prediction loss on the new adversarial samples. These loops constitute an intertwining dynamics.

---

**Algorithm 2** Federated Adversarial Learning (FAL). This is a complete version of Algorithm 1.

---

1: /\*Defining notations and parameters\*/
2:     We use $c$ to denote the client's index
3:     The training set of client $c$ is denoted as $\mathcal{S}_c = \{(x_{c,j}, y_{c,j})\}_{j=1}^{J}$
4:     Let $\mathcal{A}$ be the adversary
5:     We denote local learning rate as $\eta_{\text{local}}$
6:     We denote global learning rate as $\eta_{\text{global}}$
7:     We denote local updating iterations as $K$
8:     We denote global communication round as $T$
9:
10: /\*Initialization\*/
11:     Initialization $a(0) \in \mathbb{R}^m, U(0) \in \mathbb{R}^{d \times m}, b(0) \in \mathbb{R}^m$
12:     For $t = 0 \to T$, we iteratively run **Procedure A** then **Procedure B**
13:
14: /\* Procedure running on client side \*/
15: **procedure A**. CLIENTUPDATE$(t, c)$
16:     $\mathcal{S}_c(t) \leftarrow \emptyset$
17:     $W_c(t, 0) \leftarrow U(t)$                                    ▷ Receive global model weights update
18:     **for** $k = 0 \to K - 1$ **do**
19:         **for** $j = 1 \to J$ **do**
20:             $\widetilde{x}_{c,j}^{(t)} \leftarrow \mathcal{A}(f_{W_c(t,k)}, x_{c,j}, y_{c,j})$         ▷ Adversarial examples, $f_{W_c}$ is defined as (5)
21:             $\mathcal{S}_c(t) \leftarrow \mathcal{S}_c(t) \cup (\widetilde{x}_{c,j}^{(t)}, y_{c,j})$
22:         **end for**
23:         $W_c(t, k + 1) \leftarrow W_c(t, k) - \eta_{\text{local}} \cdot \nabla_{W_c} \mathcal{L}(f_{W_c(t,k)}, \mathcal{S}_c(t))$
24:     **end for**
25:     $\Delta U_c(t) \leftarrow W_c(t, K) - U(t)$
26:     Send $\Delta U_c(t)$ to SERVEREXECUTION
27: **end procedure**
28:
29: /\*Procedure running on server side\*/
30: **procedure B**. SERVEREXECUTION$(t)$:
31:     **for** each client $c$ **in parallel do**
32:         $\Delta U_c(t) \leftarrow$ CLIENTUPDATE$(c, t)$                    ▷ Receive local model weights update
33:         $\Delta U(t) \leftarrow \frac{1}{N} \sum_{c \in [N]} \Delta U_c(t)$
34:         $U(t + 1) \leftarrow U(t) + \eta_{\text{global}} \cdot \Delta U(t)$                    ▷ Aggregation on the server side
35:         Send $U(t + 1)$ to client $c$ for CLIENTUPDATE$(c, t)$
36:     **end for**
37: **end procedure**

---

# C  PROOF OVERVIEW

In this section we give an overview of our main result's proof. Two theorems to be used are Theorem E.3 and Theorem F.3.

## C.1  PSEUDO-NETWORK

We study the over-parameterized neural nets' well-approximated pseudo-network to learn gradient descent for over-parameterized neural nets whose weights are close to initialization. The introducing of pseudo-network makes the proof more intuitive.

To be specific, we give the definition of pseudo-network in Section D, and also state Theorem D.2 which shows the fact that the pseudo-network approximates the real network uniformly well. It can be seen that the notion of pseudo-network is used for several times in our proof.

## C.2  ONLINE GRADIENT DESCENT IN FEDERATED ADVERSARIAL LEARNING

Our federated adversarial learning algorithm is formulated in the framework of *online gradient descent*: at each local step $k$ on the client side, firstly the adversary generates adversarial samples and computes the loss function $\mathcal{L}\left(f_{W_c(t,k)}, \mathcal{S}_c(t)\right)$, then the local client learner takes the fresh loss function and update $W_c(t, k+1) = W_c(t,k) - \eta_{\text{local}} \cdot \nabla_{W_c} \mathcal{L}\left(f_{W_c(t,k)}, \mathcal{S}_c(t)\right)$. We refer our readers to Gao et al. (2019); Hazan (2016) for more details regarding online learning and online gradient descent.

Compared with the centralized setting, the key difficulties in the convergence analysis of FL are induced by multiple local updates on the client side and the updates on both local and global sides. Specifically, local updates are not the standard gradient as the centralized adversarial training when $K \geq 2$. We used $-\Delta U(t)$ in substitution of the real gradient of $U$ to update the value of $U(t)$. This brings in challenges to bound the gradient of the neural networks. Nevertheless, gradient bounding is challenging in adversarial training solely. We use gradient coupling method twice to solve this core problem: firstly we bound the difference between real gradient and FL gradient in Lemma E.4, then we bound the difference between pseudo gradient and real gradient in Lemma E.5. We show the connection of online gradient descent and federated adversarial learning in the proof of Theorem E.3.

## C.3  EXISTENCE OF ROBUST NETWORK NEAR INITIALIZATION

In Section F we show that there exists a global network $f_{U^*}$ whose weight is close to the initial value $U(0)$ and the worst-case global robust loss $\mathcal{L}_{\mathcal{A}^*}(f_{U^*})$ is sufficiently small. We show that the required width $m$ is $\text{poly}(d, (NJ/\epsilon)^{1/\gamma})$.

Suppose we are given a $\rho$-bounded adversary. For a globally $\gamma$-separable training set, in order to prove Theorem F.3, we first state Lemma F.1 which shows the existence of function $f^*$ that has "low complexity" and satisfies $f^*(\widetilde{x}_{c,j}) \approx y_{c,j}$ for all data point $(x_{c,j}, y_{c,j})$ and perturbation inputs $\widetilde{x}_{c,j} \in \mathcal{B}_2(x_{c,j}, \rho)$. Then, we state Lemma F.2 which shows the existence of a pseudo-network $g_{U^*}$ that approximates $f^*$ well. Finally, by using Theorem D.2 we show that $f_{U^*}$ approximates $g_{U^*}$ well. By combining these results, we we finish the proof of Theorem F.3.

# D  REAL APPROXIMATES PSEUDO

To make additional remark to proof sketch in Section 5, in this section, we state a tool that will be used in our proof that is related to our definition of pseudo-network. First, we recall the definition of pseudo-network.

**Definition D.1** (Pseudo-network). *Given weights $U \in \mathbb{R}^{d \times m}$, $a \in \mathbb{R}^m$ and $b \in \mathbb{R}^m$, the global neural network function $f_U : \mathbb{R}^d \to \mathbb{R}$ is defined as*

$$f_U(x) := \sum_{r=1}^{m} a_r \cdot \sigma(\langle U_r, x \rangle + b_r).$$

*Given this $f_U(x)$, we define the corresponding pseudo-network function $g_U : \mathbb{R}^d \to \mathbb{R}$ as*

$$g_U(x) := \sum_{r=1}^{m} a_r \cdot \langle U_r(t) - U_r(0), x \rangle \cdot \mathbb{1}\{\langle U_r(0), x \rangle + b_r \geq 0\}.$$

From the definition we can know that pseudo-network can be seen as a linear approximation of the two layer ReLU network we study near initialization. Next, we cite a Theorem from Zhang et al. (2020b), which gives a uniform bound of the difference between a network and its pseudo-network.

**Theorem D.2** (Uniform approximation, Theorem 5.1 in Zhang et al. (2020b)). *Let $R \geq 1$. For all $m \geq \text{poly}(d)$, with probability at least $1 - \exp(-\Omega(m^{1/3}))$ over the choice of $a(0), U(0), b(0)$, for all $U \in \mathbb{R}^{d \times m}$ such that $\|U - U(0)\|_{2,\infty} \leq R/m^{2/3}$,*

$$\sup_{x \in \mathcal{X}} |f_U(x) - g_U(x)| \leq O(R^2/m^{1/6}).$$

# E CONVERGENCE

| Section | Statement | Comment | Statements Used |
|---------|-----------|---------|-----------------|
| E.1 | Definition E.1 and E.2 | Definition | - |
| E.2 | Theorem E.3 | Convergence result | Lem. E.4, E.5, Thm. D.2 |
| E.3 | Lemma E.4 | Approximates real gradient | - |
| E.4 | Lemma E.5 | Approximates pseudo gradient | Claim E.6 |
| E.5 | Claim E.6 | Auxiliary bounding | Claim A.6 |

Table 1: List of theorems and lemmas in Section E. The main result of this section is Theorem E.3. By saying "Statements Used" we mean these statements are used in the proof in the corresponding section. For example, Lemma E.4, E.5 and Theorem D.2 are used in the proof of Theorem E.3.

## E.1 DEFINITIONS AND NOTATIONS

In Section E, we follow the notations used in Definition D.1. Since we are dealing with pseudo-network, we first introduce some additional definitions and notations regarding gradient.

**Definition E.1** (Gradient). *For a local real network $f_{W_c(t,k)}$, we denote its gradient by*

$$\nabla(f_c, t, k) := \nabla_{W_c}\mathcal{L}(f_{W_c(t,k)}, \mathcal{S}_c(t)).$$

*If the corresponding pseudo-network is $g_{W_c(t,k)}$, then we define the pseudo-network gradient as*

$$\nabla(g_c, t, k) := \nabla_{W_c}\mathcal{L}(g_{W_c(t,k)}, \mathcal{S}_c(t)).$$

Now we consider the global matrix. For convenience we write $\nabla(f, t) := \nabla_U \mathcal{L}(f_{U(t)}, \mathcal{S}(t))$ and $\nabla(g, t) := \nabla_U \mathcal{L}(g_{U(t)}, \mathcal{S}(t))$. We define the **FL gradient** as $\widetilde{\nabla}(f, t) := -\frac{1}{N}\Delta U(t)$.

**Definition E.2** (Distance). *For $U^* \in \mathbb{R}^{d \times m}$ such that $\|U^* - \widetilde{U}\|_{2,\infty} \leq R/m^{3/4}$, we define the following distance for simplicity:*

$$D_{\max} := \max_{t \in [T]} \|U(t) - \widetilde{U}\|_{2,\infty}$$

$$D_{U^*} := \|U^* - \widetilde{U}\|_{2,\infty}$$

We have $D_{U^*} = O(R/m^{3/4})$ and $\|U(t) - U^*\|_{2,\infty} \leq D_{\max} + D_{U^*}$ by using triangle inequality.

| Notation | Meaning | Satisfy |
|----------|---------|---------|
| $U(0)$ or $\widetilde{U}$ | Initialization of $U$ | $W_c(0,0) = U(0)$ |
| $U(t)$ | The value of $U$ after $t$ iterations | $D_{\max} = \max \|U(t) - \widetilde{U}\|_{2,\infty}$ |
| $U^*$ | The value of $U$ after small perturbations from $\widetilde{U}$ | $\|U^* - \widetilde{U}\|_{2,\infty} \leq R/m^{3/4}$ |

Table 2: Notations of global model weights in federated learning to be used in this section.

## E.2 CONVERGENCE RESULT

The goal of this section is to prove Theorem E.3.

**Theorem E.3** (Convergence result, formal version of Theorem 5.6). *For all $\epsilon \in (0, 1)$, for all $R \geq 1$, there exists an $M = \text{poly}(n, R, 1/\epsilon)$, such that for every $m \geq M$, for every $K \geq 1$, for every $T \geq \text{poly}(R/\epsilon)$, with probability at least $1 - \exp(-\Omega(m^{1/3}))$ over the randomness of $a(0) \in \mathbb{R}^m$, $U(0) \in \mathbb{R}^{d \times m}$, $b(0) \in \mathbb{R}^m$, if we run Algorithm 2 with setting*

$$\eta_{\text{global}} = 1/\text{poly}(NJ, R, 1/\epsilon) \quad and \quad \eta_{\text{local}} = 1/K,$$

*then for every $U^*$ such that $\|U^* - U(0)\|_{2,\infty} \leq R/m^{3/4}$, the output weights $(U(t))_{t=1}^{T}$ satisfy*

$$\frac{1}{T}\sum_{t=1}^{T}\mathcal{L}_{\mathcal{A}}\left(f_{U(t)}\right) \leq \mathcal{L}_{\mathcal{A}^*}\left(f_{U^*}\right) + \epsilon.$$

*Proof.* We set our parameters as follows:

$$M = \Omega\Big(\max\big\{(NJ)^8, (\frac{R}{\epsilon})^{12}\big\}\Big)$$

$$\eta_{\text{global}} = O(\frac{\epsilon}{Nm^{1/3} \cdot \text{poly}(R/\epsilon)})$$

$$\eta_{\text{local}} = 1/K$$

Since the loss function is 1-Lipschitz, we can first bound the norm of real net gradient:

$$\|\nabla_r(f_c, t, k)\|_2 \le |a_r| \cdot \Big(\frac{1}{J}\sum_{j=1}^{J}\sigma'(\langle W_{c,r}(t,k), x_{c,j}\rangle + b_r) \cdot \|\widetilde{x}_{c,j}\|_2\Big) \le |a_r| \le \frac{1}{m^{1/3}}. \quad (6)$$

Now we consider the pseudo-net gradient. The loss $\mathcal{L}(g_U, \mathcal{S}(t))$ is convex in $U$ due to the fact that $g$ is linear with $U$. Then we have

$$\mathcal{L}(g_{U(t)}, \mathcal{S}(t)) - \mathcal{L}(g_{U^*}, \mathcal{S}(t))$$
$$\le \langle \nabla_U \mathcal{L}(g_{U(t)}, \mathcal{S}(t)), U(t) - U^* \rangle$$
$$= \langle \widetilde{\nabla}(f, t), U(t) - U^* \rangle + \langle \nabla(f, t) - \widetilde{\nabla}(f, t), U(t) - U^* \rangle + \langle \nabla(g, t) - \nabla(f, t), U(t) - U^* \rangle$$
$$\le \alpha(t) + \beta(t) + \gamma(t)$$

where the last step follows from

$$\alpha(t) := \langle \widetilde{\nabla}(f, t), U(t) - U^* \rangle,$$
$$\beta(t) := \|\nabla(f, t) - \widetilde{\nabla}(f, t)\|_{2,1} \cdot \|U(t) - U^*\|_{2,\infty},$$
$$\gamma(t) := \|\nabla(g, t) - \nabla(f, t)\|_{2,1} \cdot \|U(t) - U^*\|_{2,\infty}.$$

Note that the FL gradient $\widetilde{\nabla}(f, t) = -\frac{1}{N}\Delta U(t)$ is the direction moved by center, in contrast, $\nabla(f, t)$ is the true gradient of function $f$. We deal with these three terms separately. As for $\alpha(t)$, we have

$$\|U(t+1) - U^*\|_F^2 = \|U(t) + \eta_{\text{global}}\Delta U(t) - U^*\|_F^2$$
$$= \|U(t) - U^*\|_F^2 - 2N\eta_{\text{global}}\alpha(t) + \eta_{\text{global}}^2\|\Delta U(t)\|_F^2$$

and by rearranging we get

$$\alpha(t) = \frac{\eta_{\text{global}}}{2N}\|\Delta U(t)\|_F^2 + \frac{1}{2N\eta_{\text{global}}} \cdot (\|U(t) - U^*\|_F^2 - \|U(t+1) - U^*\|_F^2).$$

Next, we need to upper bound $\|\Delta U(t)\|_F^2$,

$$\|\Delta U(t)\|_F^2 = \|\frac{\eta_{\text{local}}}{N}\sum_{c=1}^{N}\sum_{k=0}^{K-1}\nabla(f_c, t, k)\|_F^2$$
$$\le \frac{\eta_{\text{local}}}{N}\sum_{c=1}^{N}\sum_{k=0}^{K-1}\sum_{r=1}^{m}\|\nabla_r(f_c, t, k)\|_2^2$$
$$= \eta_{\text{local}}Km^{1/3}$$
$$= m^{1/3}. \quad (7)$$

where the last step follows from $K\eta_{\text{local}} = 1$. Then we do summation over $t$ and have

$$\sum_{t=1}^{T}\alpha(t) = \frac{\eta_{\text{global}}}{2N}\sum_{t=1}^{T}\|\Delta U(t)\|_F^2 + \frac{1}{2N\eta_{\text{global}}} \cdot \sum_{t=1}^{T}(\|U(t) - U^*\|_F^2 - \|U(t+1) - U^*\|_F^2)$$

$$= \frac{\eta_{\text{global}}}{2N}\sum_{t=1}^{T}\|\Delta U(t)\|_F^2 + \frac{1}{2N\eta_{\text{global}}} \cdot (\|U(1) - U^*\|_F^2 - \|U(T+1) - U^*\|_F^2)$$

$$\le \frac{\eta_{\text{global}}}{2N}\sum_{t=1}^{T}\|\Delta U(t)\|_F^2 + \frac{1}{2N\eta_{\text{global}}} \cdot \|U(1) - U^*\|_F^2$$

$$\lesssim \frac{\eta_{\text{global}}}{N}Tm^{1/3} + \frac{1}{N\eta_{\text{global}}}mD_{U^*}^2$$

where the last step follows from Eq. (7) and $\|\widetilde{U} - U^*\|_F^2 \le m \cdot \|\widetilde{U} - U^*\|_{2,\infty}^2 = mD_{U^*}^2$.

As for $\beta(t)$, we apply Lemma E.4 and also triangle inequality and have

$$
\begin{aligned}
\beta(t) &= \|\nabla(f,t) - \widetilde{\nabla}(f,t)\|_{2,1} \cdot \|U(t) - U^*\|_{2,\infty} \\
&\lesssim m^{2/3} \cdot \|U(t) - U^*\|_{2,\infty} \\
&\lesssim m^{2/3} \cdot (\|U(t) - \widetilde{U}\|_{2,\infty} + D_{U^*}).
\end{aligned}
$$

By using Eq. (6) we bound the size of $\|U(t) - \widetilde{U}\|_{2,\infty}$:

$$
\begin{aligned}
\|U(t) - \widetilde{U}\|_{2,\infty} &\le \eta_{\text{global}} \sum_{\tau=1}^{t} \|\Delta U(\tau)\|_{2,\infty} \\
&= \eta_{\text{global}} \sum_{\tau=1}^{t} \|\frac{\eta_{\text{local}}}{N} \sum_{c=1}^{N} \sum_{k=0}^{K-1} \nabla(f_c,t,k)\|_{2,\infty} \\
&\le \frac{\eta_{\text{global}}\eta_{\text{local}}}{N} \sum_{\tau=1}^{t} \sum_{c=1}^{N} \sum_{k=0}^{K-1} \|\nabla(f_c,t,k)\|_{2,\infty} \\
&\le \eta_{\text{global}}\eta_{\text{local}}tKm^{-1/3}
\end{aligned}
$$

and have

$$
\beta(t) \lesssim \eta_{\text{global}}\eta_{\text{local}}tKm^{1/3} + m^{2/3}D_{U^*}.
$$

Then we do summation over $t$ and have

$$
\begin{aligned}
\sum_{t=1}^{T} \beta(t) &\lesssim \sum_{t=1}^{T} (\eta_{\text{global}}\eta_{\text{local}}tKm^{1/3} + m^{2/3}D_{U^*}) \\
&\lesssim \eta_{\text{global}}\eta_{\text{local}}T^2Km^{1/3} + m^{2/3}TD_{U^*} \\
&\lesssim \eta_{\text{global}}T^2m^{1/3} + m^{2/3}TD_{U^*}.
\end{aligned}
$$

As for $\gamma(t)$, we apply Lemma E.5 and have

$$
\begin{aligned}
\gamma(t) &= \|\nabla(g,t) - \nabla(f,t)\|_{2,1} \cdot \|U(t) - U^*\|_{2,\infty} \\
&\lesssim NJm^{13/24} \cdot (\|U(t) - \widetilde{U}\|_{2,\infty} + D_{U^*}).
\end{aligned}
$$

Since $\|U(t) - \widetilde{U}\|_{2,\infty} \le \eta_{\text{global}}\eta_{\text{local}}tKm^{-1/3}$, we have

$$
\gamma(t) \lesssim \eta_{\text{global}}\eta_{\text{local}}tKNJm^{5/24} + NJm^{13/24}D_{U^*}.
$$

Then we do summation over $t$ and have

$$
\begin{aligned}
\sum_{t=1}^{T} \gamma(t) &\lesssim \sum_{t=1}^{T} \left(\eta_{\text{global}}\eta_{\text{local}}tKNJm^{5/24} + NJm^{13/24}D_{U^*}\right) \\
&\lesssim \eta_{\text{global}}\eta_{\text{local}}T^2KNJm^{5/24} + NJm^{13/24}TD_{U^*} \\
&\lesssim \eta_{\text{global}}T^2NJm^{5/24} + NJm^{13/24}TD_{U^*}.
\end{aligned}
$$

Next we put it altogether. Note that $D_{U^*} = O(\frac{R}{m^{3/4}})$, thus we obtain

$$\sum_{t=1}^{T} \mathcal{L}(g_{U(t)}, \mathcal{S}(t)) - \sum_{t=1}^{T} \mathcal{L}(g_{U^*}, \mathcal{S}(t))$$

$$\leq \sum_{t=1}^{T} \alpha(t) + \sum_{t=1}^{T} \beta(t) + \sum_{t=1}^{T} \gamma(t)$$

$$\lesssim \frac{\eta_{\text{global}}}{N} T m^{1/3} + \frac{1}{N \eta_{\text{global}}} m D_{U^*}^2 + \eta_{\text{global}} T^2 m^{1/3}$$

$$+ m^{2/3} T D_{U^*} + \eta_{\text{global}} T^2 N J m^{5/24} + N J m^{13/24} T D_{U^*}$$

$$\lesssim \frac{\eta_{\text{global}}}{N} T m^{1/3} + \frac{1}{N \eta_{\text{global}}} R^2 m^{-1/2} + \eta_{\text{global}} T^2 m^{1/3}$$

$$+ R T m^{-1/12} + \eta_{\text{global}} T^2 N J m^{5/24} + N J m^{-5/24} R T.$$

We then have

$$\frac{1}{T} \sum_{t=1}^{T} \mathcal{L}(g_{U(t)}, \mathcal{S}(t)) - \frac{1}{T} \sum_{t=1}^{T} \mathcal{L}(g_{U^*}, \mathcal{S}(t))$$

$$\lesssim \frac{\eta_{\text{global}}}{N} m^{1/3} + \frac{1}{N \eta_{\text{global}} T} R^2 m^{-1/2} + \eta_{\text{global}} T m^{1/3} + R m^{-1/12}$$

$$+ \eta_{\text{global}} T N J m^{5/24} + N J m^{-5/24} R.$$

$$\lesssim \frac{1}{N \eta_{\text{global}} T} R^2 m^{-1/2} + \eta_{\text{global}} T m^{1/3} + R m^{-1/12} + \eta_{\text{global}} T N J m^{5/24} + N J m^{-5/24} R \quad (8)$$

$$\leq O(\epsilon).$$

From Theorem D.2 we know

$$\sup_{x \in \mathcal{X}} |f_U(x) - g_U(x)| \leq O(R^2/m^{1/6}) = O(\epsilon)$$

and thus, we get

$$\frac{1}{T} \sum_{t=1}^{T} \mathcal{L}(f_{U(t)}, \mathcal{S}(t)) - \frac{1}{T} \sum_{t=1}^{T} \mathcal{L}(f_{U^*}, \mathcal{S}(t)) \leq c \cdot \epsilon \quad (9)$$

where $c > 0$ is a constant. From the definition of $\mathcal{A}^*$ we have $\mathcal{L}(f_{U^*}, S(t)) \leq \mathcal{L}_{\mathcal{A}^*}(f_{U^*})$. From the definition of loss we have $\mathcal{L}(f_{U(t)}, \mathcal{S}(t)) = \mathcal{L}_{\mathcal{A}}(f_{U(t)})$. Moreover, since Eq. (9) holds for all $\epsilon > 0$, we can replace $\frac{\epsilon}{c}$ with $\epsilon$. Thus we prove that for $\forall \epsilon > 0$,

$$\frac{1}{T} \sum_{t=1}^{T} \mathcal{L}_{\mathcal{A}}(f_{U(t)}) \leq \mathcal{L}_{\mathcal{A}^*}(f_{U^*}) + \epsilon.$$

$\square$

### E.3 APPROXIMATES REAL GLOBAL GRADIENT

The goal of this section is to prove Lemma E.4.

**Lemma E.4** (Bounding the difference between real gradient and FL gradient). *With probability at least $1 - \exp(-\Omega(m^{1/3}))$ over the randomness of $a(0) \in \mathbb{R}^m$, $U(0) \in \mathbb{R}^{d \times m}$, $b(0) \in \mathbb{R}^m$, for all iterations $t$ such that $\|U(t) - U(0)\|_{2,\infty} \leq O(m^{-15/24})$, the following holds:*

$$\|\nabla(f, t) - \widetilde{\nabla}(f, t)\|_{2,1} \leq O(m^{2/3}).$$

*Proof.* Notice that $\nabla(f, t) = \nabla_U \mathcal{L}(f_{U(t)}, \mathcal{S}(t))$ and

$$\widetilde{\nabla}(f, t) = -\frac{1}{N} \Delta U(t) = -\frac{1}{N} \sum_{c=1}^{N} \Delta U_c(t) = \frac{\eta_{\text{local}}}{N} \sum_{c=1}^{N} \sum_{k=0}^{K-1} \nabla(f_c, t, k).$$

So we have

$$
\begin{aligned}
\|\nabla(f,t) - \widetilde{\nabla}(f,t)\|_{2,1} &= \sum_{r=1}^{m} \|\nabla_r(f,t) - \widetilde{\nabla}_r(f,t)\|_2 \\
&= \frac{1}{N} \sum_{r=1}^{m} \|N \cdot \nabla_r(f,t) - \eta_{\text{local}} \sum_{c=1}^{N} \sum_{k=0}^{K-1} \nabla_r(f_c,t,k)\|_2 \\
&\leq \frac{\eta_{\text{local}}}{N} \sum_{r=1}^{m} \sum_{k=0}^{K-1} \|\frac{N \cdot \nabla_r(f,t)}{K\eta_{\text{local}}} - \sum_{c=1}^{N} \nabla_r(f_c,t,k)\|_2 \\
&= \frac{1}{NK} \sum_{r=1}^{m} \sum_{k=0}^{K-1} \|N \cdot \nabla_r(f,t) - \sum_{c=1}^{N} \nabla_r(f_c,t,k)\|_2
\end{aligned}
$$

where the last step follows from the assumption that $\eta_{\text{local}} = \frac{1}{K}$.

As for $\|N \cdot \nabla_r(f,t) - \sum_{c=1}^{N} \nabla_r(f_c,t,k)\|_2$, we have

$$
\begin{aligned}
&\|N \cdot \nabla_r(f,t) - \sum_{c=1}^{N} \nabla_r(f_c,t,k)\|_2 \\
&\leq |a_r| \cdot \Big|\Big(\frac{N}{NJ} \sum_{c=1}^{N} \sum_{j=1}^{J} \mathbb{1}\{\langle U_r(t), x_{c,j}\rangle + b_r \geq 0\} \\
&\quad - \frac{1}{J} \sum_{c=1}^{N} \sum_{j=1}^{J} \mathbb{1}\{\langle W_{c,r}(t,k), x_{c,j}\rangle + b_r \geq 0\}\Big) \cdot \|x_{c,j}\|_2\Big| \\
&\leq \frac{1}{m^{1/3}} \cdot \frac{1}{J} \sum_{c=1}^{N} \sum_{j=1}^{J} |\mathbb{1}\{\langle U_r(t), x_{c,j}\rangle + b_r \geq 0\} - \mathbb{1}\{\langle W_{c,r}(t,k), x_{c,j}\rangle + b_r \geq 0\}| \\
&\leq \frac{N}{m^{1/3}}.
\end{aligned}
$$

Then we do summation and have

$$
\begin{aligned}
\|\nabla(f,t) - \widetilde{\nabla}(f,t)\|_{2,1} &\leq \frac{1}{NK} \sum_{r=1}^{m} \sum_{k=0}^{K-1} \|N \cdot \nabla_r(f,t) - \sum_{c=1}^{N} \nabla_r(f_c,t,k)\|_2 \\
&\leq \frac{1}{NK} \sum_{r=1}^{m} \sum_{k=0}^{K-1} \frac{N}{m^{1/3}} \\
&= m^{2/3}.
\end{aligned}
$$

Thus we finish the proof. $\qquad\square$

### E.4  APPROXIMATES PSEUDO GLOBAL GRADIENT

The goal of this section is to prove Lemma E.5.

**Lemma E.5** (Bounding the difference between pseudo gradient and real gradient). *With probability at least $1 - \exp(-\Omega(m^{1/3}))$ over the randomness of $a(0) \in \mathbb{R}^m$, $U(0) \in \mathbb{R}^{d\times m}$, $b(0) \in \mathbb{R}^m$, for all iterations $t$ such that $\|U(t) - U(0)\|_{2,\infty} \leq O(m^{-15/24})$, the following holds:*

$$
\|\nabla(g,t) - \nabla(f,t)\|_{2,1} \leq O(NJm^{13/24}).
$$

*Proof.* Notice that $\nabla(g,t) = \nabla_U \mathcal{L}(g_{U(t)}, \mathcal{S}(t))$ and $\nabla(f,t) = \nabla_U \mathcal{L}(f_{U(t)}, \mathcal{S}(t))$. By Claim E.6, with the given probability we have

$$
\sum_{r=1}^{m} \mathbb{1}\{\nabla_r(g,t) \neq \nabla_r(f,t)\} \leq O(NJm^{7/8}).
$$

For indices $r \in [m]$ such that $\nabla_r(g, t) \neq \nabla_r(f, t)$, the following holds:

$$
\begin{aligned}
\|\nabla_r(g, t) - \nabla_r(f, t)\|_2 &= \|\nabla_{U,r}\mathcal{L}(g_{U(t)}, S(t)) - \nabla_{U,r}\mathcal{L}(f_{U(t)}, S(t))\|_2 \\
&\leq |a_r| \cdot \frac{1}{NJ} \cdot \sum_{c=1}^{N} \sum_{j=1}^{J} \|x_{c,j}\|_2 \cdot \big| \mathbb{1}\{\langle \widetilde{U}_r, x_{c,j}\rangle + b_r \geq 0\} \\
&\qquad - \mathbb{1}\{\langle U_r, x_{c,j}\rangle + b_r \geq 0\}\big| \\
&\leq \frac{1}{m^{1/3}} \cdot \frac{1}{NJ} \cdot \sum_{c=1}^{N} \sum_{j=1}^{J} \big| \mathbb{1}\{\langle \widetilde{U}_r, x_{c,j}\rangle + b_r \geq 0\} - \mathbb{1}\{\langle U_r, x_{c,j}\rangle + b_r \geq 0\}\big| \\
&\leq \frac{1}{m^{1/3}}.
\end{aligned}
$$

where the first step is definition, the second step follows that the loss function is 1-Lipschitz, the third step follows from $|a_r| \leq \frac{1}{m^{1/3}}$ and $\|x_{c,j}\|_2 = 1$, the last step follows from the bound of the indicator function. Thus, we do the conclusion:

$$
\begin{aligned}
\|\nabla(g, t) &- \nabla(f, t)\|_{2,1} \\
&= \sum_{r=1}^{m} \|\nabla_r(g, t) - \nabla_r(f, t)\|_2 \cdot \mathbb{1}\{\nabla_r(g, t) \neq \nabla_r(f, t)\} \\
&\leq \frac{1}{m^{1/3}} \sum_{r=1}^{m} \mathbb{1}\{\nabla_r(g, t) \neq \nabla_r(f, t)\} \\
&\leq \frac{1}{m^{1/3}} \cdot O(NJm^{7/8}) \\
&= O(NJm^{13/24})
\end{aligned}
$$

and finish the proof. $\qquad\square$

### E.5 BOUNDING AUXILIARY

**Claim E.6** (Bounding auxiliary). *With probability at least $1 - \exp(-\Omega(m^{1/3}))$ over the initialization, we have*

$$
\sum_{r=1}^{m} \mathbb{1}\{\nabla_r(g, t) \neq \nabla_r(f, t)\} \leq O(NJm^{7/8}).
$$

*Proof.* For $r \in [m]$, let $I_r := \mathbb{1}\{\nabla_r(g, t) \neq \nabla_r(f, t)\}$. By Claim A.6 we know that for each $x_{c,j}$ we have

$$
\Pr[|\langle \widetilde{W}_{c,r}, x_{c,j}\rangle + b_r| \leq m^{-15/24}] \leq O(m^{-1/8}).
$$

By putting a union bound over $c$ and $j$, we get

$$
\Pr\left[\exists c \in [N], j \in [J], \ |\langle \widetilde{W}_{c,r}, x_{c,j}\rangle + b_r| \leq m^{-15/24}\right] \leq O(NJm^{-1/8}).
$$

Since

$$
\Pr[I_r = 1] \leq \Pr[\exists c \in [N], j \in [J], \ |\langle \widetilde{W}_{c,r}, x_{c,j}\rangle + b_r| \leq m^{-15/24}],
$$

we have

$$
\Pr[I_r = 1] \leq O(NJm^{-1/8}).
$$

By applying concentration inequality on $I_r$ (independent Bernoulli) for $r \in [m]$, we have that with probability at least $1 - \exp(-\Omega(NJm^{7/8})) > 1 - \exp(-\Omega(m^{1/3}))$, the following holds:

$$
\sum_{r=1}^{m} I_r \leq O(NJm^{7/8}).
$$

Thus we finish the proof. $\qquad\square$

### E.6 FURTHER DISCUSSION

Note that in the proof of Theorem E.3 we set the hidden layer's width $m$ to be greater than $O(\epsilon^{-12})$, which seems impractical in reality: if we choose our convergence accuracy to be $10^{-2}$, the width will become $10^{24}$ which is impossible to achieve.

However, we want to claim that the "$-12$" term is not intrinsic in our theorem and proof, and we can actually further improve the lower bound of $m$ to $O((R/\epsilon)^{c_2})$ where $c_2$ is some constant between $-3$ and $-4$. To be specific, we observe from Eq. (8) that the "$-12$" term comes from $\frac{2}{3} - \frac{3}{4} = -\frac{1}{12}$, where $\frac{2}{3}$ appears in Lemma E.4 and $\frac{3}{4}$ appears in the assumption that $D_{U^*} \leq R/m^{3/4}$ in Definition E.2. As for our observations, the $\frac{2}{3}$ term is hard to improve. On the other hand, we can actually adjust the value of $D_{U^*}$ as long as we ensure

$$D_{U^*} \leq R/m^{c_3}$$

for some constant $c_3 \in (0, 1)$. When we let $c_3 \to 1$, the final result will achieve

$$O((\frac{R}{\epsilon})^3)$$

which is much more feasible in reality.

As the first work and the first step towards understanding the convergence of federated adversarial learning, the priority of our work is not achieving the tightest bounds. Instead, our main goal is to show the convergence of a general federated adversarial learning framework. Nevertheless, we will improve the bound in the final version.

# F  EXISTENCE

In this section we prove the existence of $U^*$ that is close to $U(0)$ and makes $\mathcal{L}_{\mathcal{A}^*}(f_{U^*})$ close to zero.

## F.1  TOOLS FROM PREVIOUS WORK

In order to prove our existence result, we first state two lemmas that will be used.

**Lemma F.1** (Lemma 6.2 from Zhang et al. (2020b)). *Suppose that $\|x_{c_1,j_1} - x_{c_2,j_2}\|_2 \geq \delta$ holds for each pair of two different data points $x_{c_1,j_1}, x_{c_2,j_2}$. Let $D = 24\gamma^{-1}\ln(48NJ/\epsilon)$, then there exists a polynomial $q : \mathbb{R} \to \mathbb{R}$ with degree at most $D$, size of coefficients at most $O(\gamma^{-1}2^{6D})$, such that for all $c_0 \in [N], j_0 \in [J]$ and $\widetilde{x}_{c_0,j_0} \in \mathcal{B}_2(x_{c_0,j_0}, \rho)$,*

$$\left| \sum_{c=1}^{N} \sum_{j=1}^{J} y_{c,j} \cdot q(\langle x_{c,j}, \widetilde{x}_{c_0,j_0} \rangle) - y_{c_0,j_0} \right| \leq \frac{\epsilon}{3}.$$

We let $f^*(x) := \sum_{c=1}^{N} \sum_{j=1}^{J} y_{c,j} \cdot q(\langle x_{c,j}, x \rangle)$ and have $|f^*(\widetilde{x}_{c_0,j_0}) - y_{c_0,j_0}| \leq \epsilon/3$.

**Lemma F.2** (Lemma 6.5 from Zhang et al. (2020b)). *For all $\epsilon \in (0,1)$, there exist $M = \mathrm{poly}(d, (NJ/\epsilon)^{1/\gamma})$ and $R = \mathrm{poly}((NJ/\epsilon)^{1/\gamma})$ such that for $m \geq M$, with probability at least $1 - \exp(-\Omega(\sqrt{m/NJ}))$ over the choice of $a(0) \in \mathbb{R}^m$, $U(0) \in \mathbb{R}^{d \times m}$, $b(0) \in \mathbb{R}^m$, there exists a $U^* \in \mathbb{R}^{d \times m}$ that satisfies $\|U^* - U(0)\|_{2,\infty} \leq R/m^{2/3}$ and*

$$\sup_{x \in \mathcal{X}} |g_{U^*}(x) - f^*(x)| \leq \epsilon/3.$$

## F.2  EXISTENCE RESULT

The goal of this section is to prove Theorem F.3 which is the existence result.

**Theorem F.3** (Existence, formal version of Theorem 5.2). *For all $\epsilon \in (0,1)$, there exist*

$$M_0 = \mathrm{poly}(d, (NJ/\epsilon)^{1/\gamma}) \quad and \quad R = \mathrm{poly}((NJ/\epsilon)^{1/\gamma})$$

*such that for every $m \geq M_0$, with probability at least $1 - \exp(-\Omega(m^{1/3}))$ over the randomness of $a(0) \in \mathbb{R}^m$, $U(0) \in \mathbb{R}^{d \times m}$, $b(0) \in \mathbb{R}^m$, there exists $U^* \in \mathbb{R}^{d \times m}$ that satisfies $\|U^* - U(0)\|_{2,\infty} \leq R/m^{2/3}$ and*

$$\mathcal{L}_{\mathcal{A}^*}(f_{U^*}) \leq \epsilon.$$

*Proof.* From Lemma F.1 we obtain the function $f^*$. From Lemma F.2 we know the existence of $M_0 = \mathrm{poly}(d, (NJ/\epsilon)^{1/\gamma})$ and also $R = \mathrm{poly}((NJ/\epsilon)^{1/\gamma})$. By combining these two results with Theorem D.2, we have that for all $m \geq \mathrm{poly}(d, (NJ/\epsilon)^{1/\gamma})$, with probability at least $1 - \exp(-\Omega(\sqrt{m/NJ})) - \exp(-\Omega(m^{1/3}))$, there exists a $U^* \in \mathbb{R}^{d \times m}$ that satisfies $\|U^* - U(0)\|_{2,\infty} \leq R/m^{2/3}$ and also the following properties:

- $\forall x \in \mathcal{X}, |g_{U^*}(x) - f^*(x)| \leq \epsilon/3$

- $\forall x \in \mathcal{X}, |f_{U^*}(x) - g_{U^*}(x)| \leq O(R^2/m^{1/6})$

We consider the loss function. For all $c \in [N]$, $j \in [J]$ and $\widetilde{x}_{c,j} \in \mathcal{B}(x_{c,j}, \rho)$, we have

$$
\begin{aligned}
\ell(f_{U^*}(\widetilde{x}_{c,j}), y_{c,j}) &\leq |f_{U^*}(\widetilde{x}_{c,j}) - y_{c,j}| \\
&\leq |f_{U^*}(\widetilde{x}_{c,j}) - g_{U^*}(\widetilde{x}_{c,j})| + |g_{U^*}(\widetilde{x}_{c,j}) - f^*(\widetilde{x}_{c,j})| + |f^*(\widetilde{x}_{c,j}) - y_{c,j}| \\
&\leq O(R^2/m^{1/6}) + \frac{\epsilon}{3} + \frac{\epsilon}{3} \\
&\leq \epsilon,
\end{aligned}
$$

Thus, we have that $\mathcal{L}_{\mathcal{A}^*}(f_{U^*}) = \frac{1}{NJ} \sum_{c=1}^N \sum_{j=1}^J \max \ell \left( f_{U^*}(x_{c,j}^*), y_{c,j} \right) \leq \epsilon$. Furthermore, since the $m$ we consider satisfies $m \geq \Omega((NJ)^{1/\gamma})$, the holding probability is at least

$$1 - \exp(-\Omega(\sqrt{m/NJ})) - \exp(-\Omega(m^{1/3})) = 1 - \exp(-\Omega(m^{1/3})).$$

Thus, we finish the proof of this theorem.

$\square$

