# OpenReview forum: "Provable Federated Adversarial Learning via Min-max Optimization"
_ICLR.cc/2022/Conference — ICLR 2022 Submitted_

### Official Review · Reviewer_vGrK · 2021-10-28

**Correctness:** 3
**Technical Novelty And Significance:** 2
**Empirical Novelty And Significance:** Not applicable
**Recommendation:** 3
**Confidence:** 3

**Main Review:**

Strengths: combining adversarial robustness with federated learning is definitely a new setting, and has the potential of training robust models collaboratively against perturbation. This paper aims to study FedAvg theoretically in this setting, and it gives theoretical results for the convergence, by controlling the noise from local updates, from the two-layer network, and from the adversarial perturbation.

Weakness: one of my major concerns is the novelty against Zhang et al 2020b. Specifically,

1) Def 3.1, 3,2 are the same as Sec 3.2 from Zhang et al. (They are not definitions per se but Assumptions)
2) Def 3.3 is the same as Def 3.2 from Zhang et al.
3) Def 3.4 is the same as Def 3.4 from Zhang et al.
4) Def 3.5 is the same as Def 3.1 from Zhang et al.
5) Def 3.6 resembles Def 3.3 from Zhang et al.
6) Thm 4.1 resembles Thm 4.1 from Zhang et al.

The techniques such as pseudo-network is also from Zhang et al.

The improvement is not strong either. Since Zhang et al. proposed linear pseudo-network to approximate the two-layer over-parametrized ReLU, using FL gradient to approximate the gradient is nothing but adding another approximation error term.

Given that the theoretical contribution is not quite strong, I would expect the authors to conduct experiments in FL adversarial training, to verify their theory. However, no experiment is provided. The reason might be because the theory analyzed does not exactly match the common experimental setting in FL, e.g., two-layer neural networks, polynomial hidden units, convex Lipschitz loss, and gamma-separability.
I doubt whether such theory could be of any use to the FL community.

In fact, many adversarial attacks are not small perturbation, e.g., adversarial patch:

[1] Brown et al, Adversarial patch, NIPS 2017.

However, the current theory requires very small perturbation in order to control the approximation error (Table 2, Appendix E).


**Summary Of The Paper:**

This paper is a direct follow-up of Zhang et al 2020b. With assumptions including overparametrized two-layer ReLU network, normalized dataset, gamma-separability, and Lipschitz convex loss, it proves the convergence of FedAvg under adversarial perturbation. These assumptions are easy adaptations from Zhang et al 2020b.


--post rebuttal--

I would like to thank the authors for the response. However, after reading the author response my opinion remains the same, as the authors acknowledge that this work is an extension of Zhang et al 2020b and there is no experiment after the revision. My main concern is still the novelty. I would encourage the authors to work on the future direction for empirically verifying their general framework.



**Summary Of The Review:**

Given the above novelty concern and the lack of experiments, I would recommend rejection. I would suggest the authors rethink the practicability of the assumptions considered, rather than directly borrow these assumptions from existing literature. It would be much more interesting if the authors could give some experiments to demonstrate the proposed algorithm.

---

> ### Author Response · Authors · 2021-11-22
> **Reply to R5**
>
> We sincerely thank you for your valuable feedback. Zhang et al 2020b gave a theoretical proof of the convergence of adversarial training in the centralized domain, while our work extends their result: we build a general Federated Adversarial Learning framework, and prove its convergence. One of the main difficulties (which does not appear in the centralized setting) we encountered in the proof is the model not updating in the gradient direction due to the multiple local steps in FL. We solved it by using the gradient coupling method twice in Lemma 5.4 and 5.5.
>
> As for experiments, it might be challenging to design suitable experiments to demonstrate our general framework. As an interesting future direction, we will consider conducting experiments by narrowing down a certain algorithm for $\mathcal{A}$.

---

### Official Review · Reviewer_Pwgr · 2021-11-01

**Correctness:** 3
**Technical Novelty And Significance:** 2
**Empirical Novelty And Significance:** 2
**Recommendation:** 3
**Confidence:** 3

**Main Review:**


1) It is unclear what are the technical difficulties to extend classical centralized adversarial training to the federated setting. Given current researches on adversarial learning, federated learning and deep learning theory, what are the contributions of this submission?  What are the gaps that this paper addresses?

2) There is a lack of experiments, which makes the proposed methods unconvincing. Can the proposed framework really enhance the robustness of the federated learning? Have the data heterogeneity issue really get solved? And so on.

3) It would be very helpful that the authors give some concrete application examples in federated learning where a specific data generating algorithm  \mathcal{A} can help to fix a certain type of robust concern, by both theory and experiments.

I would like to see these concerns can be addressed by the authors' feedback.

**Summary Of The Paper:**

This paper proposes a federated adversarial learning framework where clients generate adversarial data and do local updates while the server aggregate the local models to do global updates.

**Summary Of The Review:**

I tend to vote for a rejection due to the lack of novelty, clear theoretical justifications and experiments.

---

> ### Author Response · Authors · 2021-11-22
> **Reply to R4**
>
> We sincerely thank you for your valuable feedback. We have addressed your concerns as follows.
> 1. Technical difficulties and contributions: Compared with the centralized setting, the main difficulties in the convergence analysis of FL are induced by multiple local updates and non-IID clients. To solve the problem, we use the gradient coupling method twice in Lemma 5.4 and 5.5. We think our method of handling gradients will be of help in theoretically analyzing FL. To the best of our knowledge, our work is the first to study the convergence theory of adversarial training in FL. Our work provides theoretical implications to combine the robust benefits of adversarial training and the privacy preserving benefits of federated learning.
> 2. Experiment and concrete examples: Thank you for the suggestions. Our main goal in this work is to build a general framework for federated adversarial learning analysis and prove its convergence. Also, without any specification, it might be challenging to design suitable experiments to demonstrate the general framework. As an interesting future direction, if we narrow down a certain algorithm for $\mathcal{A}$, it will be helpful to conduct experiments.

---

### Official Review · Reviewer_Gk6a · 2021-11-01

**Correctness:** 3
**Technical Novelty And Significance:** 3
**Empirical Novelty And Significance:** Not applicable
**Recommendation:** 5
**Confidence:** 3

**Main Review:**

Pros:

1. This paper is well written and organized, which allows a clear read.
2. This work is well motivated by the challenge of applying adversarial training into federated learning.
3. This paper provides a framework to analyze federated adversarial training in over-parameterized neural networks.

Cons:

1. The first reason for harder to analyze the convergence of federated adversarial training seems to be not unique in the federated learning paradigm, and (in my opinion) the second reason is the cause of the former. Thus, it may be better to consider the Non-IID setting in federated learning as one important research focus when analyzing the plain combination of adversarial training with federated learning or to give more insights about the challenge of adversarial training with the Non-IID setting.
2. Although this paper provides the comprehensive proof of theoretical convergence guarantee for over-parameterized ReLU network on the presented federated adversarial training, it is limited in the term of showing other insights for the unique challenge of adversarial training will meet in federated learning, which may be more significant and interesting for other future research.

Minor:

1. The two reasons in the abstract seem to be not consistent with the "involved challenges" in the latter of the introduction, are they refer to the same focus?

**Summary Of The Paper:**

This paper studies the provable property of applying adversarial training into federated learning from the theoretical perspective. Specifically, the authors provide a framework for analyzing federated adversarial training and present the convergence analysis in the over-parameterized regime, the main results theoretically show that the minimal value of loss function in this learning paradigm can converge to $\epsilon$ small under certain circumstances.

**Summary Of The Review:**

This paper provides convergence analysis for federated adversarial training, which theoretically shows the feasibility of applying adversarial training to federated learning. However, it may be not easy to see some in-depth insights or huge novelty in the current submission.

---

> ### Author Response · Authors · 2021-11-22
> **Reply to R3**
>
> Thank you for your valuable feedback. We have addressed your concerns as follows.
> 1. Non-IID setting: We agree that non-IID is an important setting in FL, and actually we have mentioned that our analysis framework is feasible for non-IID clients (in the first bullet point at the end of page 2, and at the end of Section 3.2). It is worth noting that our analysis does not require assumptions on data distribution. We only assume that the data are well-separated.
> 2. Challenges and techniques: As we stated in our introduction and method section, our FAL algorithm has an inner loop in the local updating cycle that generates adversarial examples for training and an outer loop in which the parameters are updated subject to a gradient descent step. Since in FL the update direction after multi-local steps does not equal the gradient, the update in the global model is no longer determined by the gradient directions directly. So we define FL gradient right after Definition 5.3, and use the gradient coupling method twice (Lemma 5.4 and 5.5) in order to prove the bound.
> 3. Minor question: We are thankful for the minor you mentioned, we would like to clarity that there shall be three challenges: (i) the complexity of min-max optimization, (ii) model not updating in the gradient direction due to the multi-local updates on the client-side before aggregation and (iii)  inter-client heterogeneity issue. We have corrected this in our revision.

---

### Official Review · Reviewer_KhJu · 2021-11-01

**Correctness:** 4
**Technical Novelty And Significance:** 2
**Empirical Novelty And Significance:** Not applicable
**Recommendation:** 5
**Confidence:** 3

**Main Review:**

Strength:
1\ This paper provides rigorous analysis and all the details are well written.
2\ Adversarial learning and federated learning are both becoming more and more important in ML area. This paper is the first works to give theoretical guarantees on AFL which I believe has some significance.
3\ Their proposed AFL framework also shows many promising directions for future analysis.

Weakness:
1\  As the authors claimed in the paper, their analysis is not tight. They argue that their main focus is to give the first step for AFL convergence analysis.
2\ Their technical contributions, despite rigorous analysis, is limited:
a. They use many assumptions include lie overparameterization, separability, and bounded adversarial samples. Although I understand this is somehow standard in much previous analysis, I still think it simplifies a lot of proofs.
b. Many of their proofs are built upon previous results. For example, the first part of their analysis -- the existence of small robust loss, is mostly a small modification from existing methods (as the authors show in the appendix. F) and has little relation with FL. In the second part, although they emphasize their analysis on the newly defined FL gradient, to me it is a very natural definition. And the analysis to bound those differences is also expected.

Question:
In Definition B.4 ($\gamma$-seperability), it seems to me in order to satisfy $\gamma \leq \delta(\delta-2\rho)$, we requires $\rho < \delta/2 < 1/4$.  And I wonder will this assumption simplifies the analysis a lot? Can you give more justification for this assumption？ What if you have a weaker assumption like allows the adversary to be even stronger, is it still possible to prove the convergence?


**Summary Of The Paper:**

This paper proposes a federated adversarial learning (FAL) framework with strong theoretical guarantees. Compared to the centralized model, the federated model allows each local client to generate the adversarial samples and updates the gradient themselves for several iterations and then communicate to the centralized model for global updating.  This is the first work that gives convergence guarantees for FAL.

Their technical analysis mainly involves two parts In the first part, they utilize the overparameterization and separability assumption to ensure the initialized model is close to some model U^* which can achieve the small robust loss. Then in the second part, by using such property on the initialized model and by bounding the difference between real gradient and the FL gradient, they are able to show the convergence. As the authors stated, this FL gradient is a new gradient they use to tackle the difference between global and local updates.

**Summary Of The Review:**

Overall I think the author proposes a useful framework with rigorous analysis but I am somehow concerned with the technical novelty of this paper. I think it would be better if the author can get a higher bound and make more efforts to relax some assumptions (or give more discussion on the necessity/difficulty of those assumptions)

---

> ### Author Response · Authors · 2021-11-22
> **Reply to R2**
>
> Thank you for your informative and valuable feedback. We have addressed your concerns as follows.
> 1. Tightness of our analysis: As the first step towards theoretically understanding the convergence of federated adversarial learning, our work did not aim at achieving tight bounds. Instead, our main message is to show the convergence of a general federated adversarial learning framework. Moreover, we give a tightness analysis in Appendix E.6. We believe our analysis will shed light on the theoretical insights of FAL and motivate follow-up works.
> 2. Assumptions and Technical contribution: We agree that some assumptions are a little bit strict. However, most of our assumptions are widely used in the theoretical proof of convergence, such as [1], [2] and [3]. Our main innovation lies in that we provide a general FAL framework and we have detailed our techniques at the end of the introduction part in page 2. To be specific, we design our FAL as an algorithm that has an inner loop in the local updating cycle which generates adversarial examples for training, and an outer loop in which the parameters are updated subject to a gradient descent step. Since in FL the update direction after multi-local steps does not equal the gradient, the update in the global model is no longer determined by the gradient directions directly. So we define FL gradient right after Definition 5.3, and use the gradient coupling method twice (Lemma 5.4 and 5.5) in order to prove the bound.
> 3. Thank you for your great question. Actually the $\gamma \leq \delta (\delta - 2 \rho)$ assumption does not simplify our proof. Instead we can revise this Definition B.4 and say $\delta$-separability, since $\gamma$ is only a constant that is stated in our main Theorem 4.1, i.e., $R = poly((NJ / \epsilon)^{1 / \gamma})$. This $\delta$-separability saying is only for convenience. Also, we do rely on $\rho < \delta/2 < 1/4$ in our proof. We think this assumption is reasonable for adversarial training. Nevertheless, we also agree that it is an interesting future direction to remove this assumption and allow for stronger adversaries.
>
> [1] Allen-Zhu, Zeyuan, Yuanzhi Li, and Zhao Song. "A convergence theory for deep learning via over-parameterization." International Conference on Machine Learning. PMLR, 2019.
>
> [2] Allen-Zhu, Zeyuan, Yuanzhi Li, and Yingyu Liang. "Learning and generalization in overparameterized neural networks, going beyond two layers." Advances in neural information processing systems (2019).
>
> [3] Zhang, Yi, et al. "Over-parameterized adversarial training: An analysis overcoming the curse of dimensionality." arXiv preprint arXiv:2002.06668 (2020).

---

### Official Review · Reviewer_e6rQ · 2021-11-02

**Correctness:** 4
**Technical Novelty And Significance:** 2
**Empirical Novelty And Significance:** Not applicable
**Recommendation:** 3
**Confidence:** 3

**Main Review:**

Strengths:
* The robustness of neural networks is an important problem and how the distributed learning will converge under adversarially robust training is also crucial in practice.
* The authors have clarified most assumptions and conclusions.
* The proposed analysis method, pseudo gradient based on a pseudo-network, is interesting, which may be inspiring for other theoretic work.

Weakness
* The challenges caused by the interaction of the federated optimization and the min-max optimization are not clear enough. It is not specific enough to mention that local update steps and global communication are novel settings. The authors may need to elaborate on why the integration is hard in terms of convergence analysis. Specifically, how the problem is different from traditional federated with distributional shift [A] if the adversarial noise is simply treated as a bounded noise?
* The discussion of related work on federated convergence seems missing, either in the introduction or related work sections. For example, both [A] and [B] theoretically analyze the federated learning under noise.
* The technical implications of the main theorem (4.1) are not elaborated, either in Section 4 or later. The authors only vaguely mention that controlling local/global learning rates will be helpful for convergence (in Section 6) but not clear how. Actually, I believe it is folk knowledge that decreasing the learning rate is important merely for FedAvg [C]. Lacking useful implications apparently lowers the value of the work in the community of ICLR.

References:
* [A] Reisizadeh, A., Farnia, F., Pedarsani, R., & Jadbabaie, A. (2020). Robust Federated Learning: The Case of Affine Distribution Shifts. NeurIPS.
* [B] Yin, D., Chen, Y., Kannan, R., & Bartlett, P. (2018). Byzantine-Robust Distributed Learning: Towards Optimal Statistical Rates. ICML
* [C] Li, X., Huang, K., Yang, W., Wang, S., & Zhang, Z. (2020). On the Convergence of FedAvg on Non-IID Data. ICLR

**Summary Of The Paper:**

The paper formulate a general form of federated adversarial learning (FAL), based on which the authors theoretically analyze the convergence of FAL. They concluded that convergence is possible with chosen learning rate and communication round for distribution-agnostic clients.

**Summary Of The Review:**

The studied problem is interesting and important. However, the related work is not well discussed and the conclusion is weak for me. The authors should try to clarify and establish the novelty of the problem contradicting related robust federated learning.

---

> ### Author Response · Authors · 2021-11-22
> **Reply to R1**
>
> Thank you for your valuable comments. We have addressed them as follows.
> 1. Local update steps and global communication is a common setting in FL, which induces the challenge that the model does not update in the gradient direction, thus it is hard to bound the updating term. Different from related work [A] that was built on several assumptions on the loss functions without a direct connection to neural networks, we would like to point out that our method is analysis on neural networks. Specifically, our analysis is based on over-parameterized neural networks. Nevertheless, we have added [A] to the related work section and illustrated the differences in our revision.
> 2. Thanks for pointing us to [B]. This work develops distributed optimization algorithms that are provably robust against arbitrary and potentially adversarial behavior in distributed computing systems. [B] mainly focus on achieving optimal statistical performance, while we mainly focus on building a general FAL framework and prove its convergence. We have also added [B] to the related work section and illustrated the differences in our revision.
> 3. Technical implications: In our main theorem, we provided implications on how to control learning rates and to improve the convergence of FAL. Moreover, it is noteworthy that our work is different from FedAVG in that we handle online min-max optimization.
>
> References:
>
> [A] Reisizadeh, A., Farnia, F., Pedarsani, R., & Jadbabaie, A. (2020). Robust Federated Learning: The Case of Affine Distribution Shifts. NeurIPS.
>
> [B] Yin, D., Chen, Y., Kannan, R., & Bartlett, P. (2018). Byzantine-Robust Distributed Learning: Towards Optimal Statistical Rates. ICML

---

### Author Response · Authors · 2021-11-22
**Thank you**

We would like to thank all the reviewers for their valuable comments and suggestions. We have addressed the comments and updated our manuscript correspondingly. Following the suggestions, we have enriched our related work discussion, and also corrected the minor mistake in our revision which is highlighted in red.

---

### Decision · Program_Chairs · 2022-01-20

**Decision:**

Reject

**Comment:**

The reviewers had a number of concerns which seem to remain after the authors response. In particular, the reviewers were concerned about the validity of the paper's assumptions in real-world applications and lack of experimental results. Also, while the reviewers acknowledge the novelty in technical contributions, they suggested that the authors explain more clearly how the results of this paper are distinguishable from prior art.